# Deflated Dynamics Value Iteration

**Jongmin Lee**                                                        *dlwhd2000@snu.ac.kr*
*Seoul National University*

**Amin Rakhsha**                                                       *aminr@cs.toronto.edu*
*Department of Computer Science, University of Toronto*
*Vector Institute*

**Ernest K. Ryu**                                                      *eryu@math.ucla.edu*
*University of California, Los Angeles*

**Amir-massoud Farahmand**                          *amir-massoud.farahmand@polymtl.ca*
*Polytechnique Montréal*
*Mila - Quebec AI Institute*
*University of Toronto*

**Reviewed on OpenReview:** *https://openreview.net/forum?id=IbQTE24aZw*

## Abstract

The Value Iteration (VI) algorithm is an iterative procedure to compute the value function of a Markov decision process, and is the basis of many reinforcement learning (RL) algorithms as well. As the error convergence rate of VI as a function of iteration $k$ is $O(\gamma^k)$, it is slow when the discount factor $\gamma$ is close to 1. To accelerate the computation of the value function, we propose Deflated Dynamics Value Iteration (DDVI). DDVI uses matrix splitting and matrix deflation techniques to effectively remove (deflate) the top $s$ dominant eigen-structure of the transition matrix $\mathcal{P}^\pi$. We prove that this leads to a $\tilde{O}(\gamma^k|\lambda_{s+1}|^k)$ convergence rate, where $\lambda_{s+1}$ is the $(s+1)$-th largest eigenvalue of the dynamics matrix. We also extend DDVI to the RL setting and present Deflated Dynamics Temporal Difference (DDTD) algorithm. We empirically show the effectiveness of the proposed algorithms.

## 1 Introduction

Computing the value function $V^\pi$ for a policy $\pi$ or the optimal value function $V^\star$ is an integral step of many planning and reinforcement learning (RL) algorithms. Value Iteration (VI) is a fundamental dynamic programming algorithm for computing the value functions, and its approximate and sample-based variants, such as Temporal Different Learning (Sutton, 1988), Fitted Value Iteration (Ernst et al., 2005; Munos & Szepesvári, 2008), Deep Q-Network (Mnih et al., 2015), are the workhorses of modern RL algorithms (Bertsekas & Tsitsiklis, 1996; Sutton & Barto, 2018; Szepesvári, 2010; Meyn, 2022).

The VI algorithm, however, can be slow for problems with long effective planning horizon problems, when the agent has to look far into future in order to make good decisions. Within the discounted Markov Decision Processes formalism, the discount factor $\gamma$ determines the effective planning horizon, with $\gamma$ closer to 1 corresponding to longer planning horizon. The error convergence rate of the conventional VI as a function of the iteration $k$ is $O(\gamma^k)$, which is slow when $\gamma$ is close to 1.

Recently, there has been a growing body of research that explores the application of acceleration techniques of other areas of applied math to planning and RL: Geist & Scherrer (2018); Sun et al. (2021); Ermis & Yang (2020); Park et al. (2022); Ermis et al. (2021); Shi et al. (2019) applies Anderson acceleration of fixed-point iterations, Lee & Ryu (2023) applies Anchor acceleration of minimax optimization, Vieillard et al. (2020); Goyal & Grand-Clément (2022); Grand-Clément (2021); Bowen et al. (2021); Akian et al. (2022) applies

Nesterov acceleration of convex optimization, and Farahmand & Ghavamzadeh (2021); Bedaywi et al. (2024) borrow ideas from control theory/engineering, specifically PID controllers, to accelerate planning and RL algorithms.

We introduce a novel approach to accelerate VI based on modification of the eigenvalues of the transition dynamics, which are closely related to the convergence rate of VI. To see this connection, consider the policy evaluation problem where the goal is to find the value function $V^\pi$ for a given policy $\pi$. VI starts from an arbitrary $V^0$ and iteratively sets $V^{k+1} \leftarrow r^\pi + \gamma \mathcal{P}^\pi V^k$, where $r^\pi$ and $\mathcal{P}^\pi$ are the reward vector and the transition matrix of policy $\pi$, respectively. If $V^0 = 0$, the error vector/function at iteration $k$ is $V^\pi - V^k = \sum_{i=k}^\infty (\gamma \mathcal{P}^\pi)^i r^\pi$. Let us take a closer look at this difference.

For simplicity, assume that $\mathcal{P}^\pi$ is a diagonalizable matrix, so we can write $\mathcal{P}^\pi = UDU^{-1}$ with $U$ consisting of the (right) eigenvectors of $\mathcal{P}^\pi$ and $D$ being the diagonal matrix $\operatorname{diag}(\lambda_1, \dots, \lambda_n)$ with $1 = |\lambda_1| \geq \dots \geq |\lambda_n|$. Since $(\mathcal{P}^\pi)^i = UD^iU^{-1}$, after some manipulations, we see that

$$V^\pi - V^k = U \begin{bmatrix} \frac{(\gamma\lambda_1)^k}{1-\gamma\lambda_1} & 0 & \dots & 0 \\ 0 & \frac{(\gamma\lambda_2)^k}{1-\gamma\lambda_2} & \dots & 0 \\ \vdots & \dots & \ddots & 0 \\ 0 & \dots & 0 & \frac{(\gamma\lambda_n)^k}{1-\gamma\lambda_n} \end{bmatrix} U^{-1} r^\pi.$$

The diagonal terms are of the form $(\gamma\lambda_i)^k$, so they all converge to zero. The dominant term is $(\gamma\lambda_1)^k$, which corresponds to the largest eigenvalue. As the largest eigenvalue $\lambda_1$ of the stochastic matrix $\mathcal{P}^\pi$ is 1, this leads to the dominant behaviour of $O(\gamma^k)$. This is the same rate that we also get from the cruder norm-based contraction mapping analysis. The second dominant term behaves as $O((\gamma|\lambda_2|)^k)$, and so on.

If we could somehow remove from $\mathcal{P}^\pi$ the subspace corresponding to the top $s$ eigen-structure with eigenvalues $\lambda_1, \dots, \lambda_s$, the dominant behaviour of the new procedure would be $O((\gamma|\lambda_{s+1}|)^k)$, which can be much faster than $O(\gamma^k)$ of the conventional VI. Although this is perhaps too good to seem feasible, this is exactly what the proposed *Deflated Dynamics Value Iteration* (DDVI) algorithm achieves. Even if $\mathcal{P}^\pi$ is not a diagonalizable matrix, DDVI works. DDVI is based on two main ideas.

The first idea is the *deflation technique*, well-studied in linear algebra (see Section 4.2 of Saad 2011), that allows us to remove large eigenvalues of $\mathcal{P}^\pi$. This is done by subtracting a matrix $E$ from $\mathcal{P}^\pi$. This gives us a "deflated dynamics" $\mathcal{P}^\pi - E$ that does not have eigenvalues $\lambda_1, \dots, \lambda_s$. The second idea is based on *matrix splitting* (see Section 11.2 of Golub & Van Loan 2013), which allows us to use the modified dynamics $\mathcal{P}^\pi - E$ to define a VI-like iterative procedure and still converge to the same solution $V^\pi$ to which the conventional VI converges.

Deflation has been applied to various algorithms such as conjugate gradient algorithm (Saad et al., 2000), principal component analysis (Mackey, 2008), generalized minimal residual method (Morgan, 1995), and nonlinear fixed point iteration (Shroff & Keller, 1993) for improvement of convergence. While some prior work in accelerated planning can be considered as special cases of deflation (Bertsekas, 1995; White, 1963), to the best of our knowledge, this is the first application of the deflation technique that eliminates multiple eigenvalues in the context of RL. On the other hand, multiple planning algorithms have been introduced based on the general idea of matrix splitting (Hastings, 1968; Kushner & Kleinman, 1968; 1971; Reetz, 1973; Porteus, 1975; Rakhsha et al., 2022), though with a different splitting than this work.

After a review of relevant background in Section 2, we present DDVI and prove its convergence rate for the Policy Evaluation (PE) problem in Section 3. Next, in Section 4, we discuss the practical computation of the deflation matrix. In Section 5, we explain how DDVI can be extended to its sample-based variant and introduce the Deflated Dynamics Temporal Difference (DDTD) algorithm. Finally, in Section 6, we empirically evaluate the proposed methods and show their practical feasibility.

## 2 Background

We first briefly review basic definitions and concepts of Markov Decision Processes (MDP) and Reinforcement Learning (RL) (Bertsekas & Tsitsiklis, 1996; Sutton & Barto, 2018; Szepesvári, 2010; Meyn, 2022). We then describe the Power Iteration and QR Iterations, which can be used to compute the eigenvalues of a matrix.

For a measurable set $\Omega$, we denote $\mathcal{M}(\Omega)$ as the space of probability distributions over set $\Omega$. We use $\mathcal{F}(\Omega)$ to denote the space of bounded measurable real-valued functions over $\Omega$. For a matrix $A$, we use spec(A) to denote its spectrum (the set of eigenvalues), $\rho(A)$ to denote its spectral radius (the maximum of the absolute value of eigenvalues), and $\|\cdot\|$ to denote its $l^\infty$ norm.

**Markov Decision Process.** The discounted MDP is defined by the tuple $(\mathcal{X}, \mathcal{A}, \mathcal{P}, \mathcal{R}, \gamma)$, where $\mathcal{X}$ is the state space, $\mathcal{A}$ is the action space, $\mathcal{P} \colon \mathcal{X} \times \mathcal{A} \to \mathcal{M}(\mathcal{X})$ is the transition probability kernel, $\mathcal{R} \colon \mathcal{X} \times \mathcal{A} \to \mathcal{M}(\mathbb{R})$ is the reward kernel, and $\gamma \in [0, 1)$ is the discount factor. In this work, we assume that the MDP has a finite number of states. We use $r \colon \mathcal{X} \times \mathcal{A} \to \mathbb{R}$ to denote the expected reward at a given state-action pair. Denote $\pi \colon \mathcal{X} \to \mathcal{M}(\mathcal{A})$ for a policy. We define the reward function for a policy $\pi$ as $r^\pi(x) = \mathbf{E}_{a \sim \pi(\cdot|x)}[r(x, a)]$. The transition kernel of following policy $\pi$ is denoted by $\mathcal{P}^\pi \colon \mathcal{X} \to \mathcal{M}(\mathcal{X})$ and is $\mathcal{P}^\pi(x' \mid x) = \sum_{a \in \mathcal{A}} \pi(a \mid x)\mathcal{P}(x' \mid x, a)$ (or an integral over actions, if the action space is continuous).

The value function for a policy $\pi$ is $V^\pi(x) = \mathbf{E}_\pi[\sum_{t=0}^\infty \gamma^t r(x_t, a_t) \mid x_0 = x]$ where $\mathbf{E}_\pi$ denotes the expected value over all trajectories $(x_0, a_0, x_1, a_1, \dots)$ induced by $\mathcal{P}$ and $\pi$. We say that $V^\star$ is optimal value functions if $V^\star = \sup_\pi V^\pi$. We say $\pi^\star$ is an optimal policy if $\pi^\star \in \text{Arg}\max_\pi V^\pi$.

The Bellman operator $T^\pi$ for policy $\pi$ is defined as the mapping that takes $V \in \mathcal{F}(\mathcal{X})$ and returns a new function such that its value at state $x$ is $(T^\pi V)(x) = \mathbf{E}_{a \sim \pi(\cdot|x), x' \sim \mathcal{P}(\cdot|x,a)}[r(x, a) + \gamma V(x')]$. The Bellman optimality operator $T^\star$ is defined as $(T^\star V)(x) = \sup_{a \in \mathcal{A}} \{r(x, a) + \gamma \mathbf{E}_{x' \sim \mathcal{P}(\cdot \mid x, a)}[V(x')]\}$. The value functions $V^\pi$ and $V^\star$ are the fixed points of the Bellman operators, that is, $V^\pi = T^\pi V^\pi$ and $V^\star = T^\pi V^\star$.

**Value Iteration.** The Value Iteration algorithm is one of the main methods in dynamic programming and planning for computing the value function $V^\pi$ of a policy (the Policy Evaluation (PE) problem), or the optimal value function $V^\star$ (the Control problem). It is iteratively defined as

$$V^{k+1} \leftarrow \begin{cases} T^\pi V^k & \text{(Policy Evaluation)} \\ T^\star V^k & \text{(Control)}, \end{cases}$$

where $V^0$ is the initial function. For discounted MDPs where $\gamma < 1$, the Bellman operators are contractions, so by the Banach fixed-point theorem (Banach, 1922; Hunter & Nachtergaele, 2001; Hillen, 2023), the VI converges to the unique fixed points, which are $V^\pi$ or $V^\star$, with the convergence rate of $O(\gamma^k)$.

Let us now recall some concepts and methods from (numerical) linear algebra, see Golub & Van Loan (2013) for more detail. For any $A \in \mathbb{R}^{n \times n}$ (not necessarily symmetric nor diagonalizable) and for any matrix norm $\|\cdot\|$, Gelfand's formula states that spectral radius $\rho(A)$ satisfies $\rho(A) = \lim_{k \to \infty} \|A^k\|^{1/k}$. Hence, $\|A^k\| = O((\rho(A) + \epsilon)^k)$ for any $\epsilon > 0$, which we denote by $\|A^k\| = \tilde{O}((\rho(A))^k)$. Furthermore, if $A$ is diagonalizable, then $\|A^k\| = O(\rho(A)^k)$.

**Power Iteration.** Powers of $A$ can be used to compute eigenvalues of $A$. The Power Iteration starts with an initial vector $b^0 \in \mathbb{R}^n$ and for $k = 0, 1, 2, \dots$ computes

$$b^{k+1} = \frac{Ab^k}{\|Ab^k\|_2}.$$

If $\lambda_1$ is the eigenvalue such that $|\lambda_1| = \rho(A)$ and the other eigenvalues $\lambda_2, \dots, \lambda_n$ have strictly smaller magnitude, then $(b^k)^\top Ab^k \to \lambda_1$ for almost all starting points $b^0$ (Section 7.3.1 of Golub & Van Loan 2013).

**QR iteration.** The QR Iteration (or Orthogonal Iteration) algorithm (Golub & Van Loan, 2013, Sections 7.3 and 8.2) is a generalization of the Power Iteration for finding multiple eigenvalues. For any $A \in \mathbb{C}^{n \times n}$, let

$U^0 \in \mathbb{C}^{n \times s}$ have orthonormal columns and perform

$$Z^{k+1} = AU^k$$
$$U^{k+1}R^{k+1} = Z^{k+1} \quad \text{(QR factorization)}$$

for $k = 0, 1, \ldots$. If $|\lambda_s| > |\lambda_{s+1}|$, the columns of $U^k$ converge to an orthonormal basis for the dominant $s$-dimensional invariant subspace associated with the eigenvalues $\lambda_1, \ldots, \lambda_s$ and diagonal entries of $(U^s)^\top AU^s$ converge to $\lambda_1, \ldots, \lambda_s$ for almost all starting $U^0$ (Golub & Van Loan, 2013, Theorem 7.3.1). Each QR iteration needs $O(n^2 s)$ flops for its matrix multiplication, which is the dominant part of the computation, and $O(ns^2)$ flops for the QR factorization.

## 3 Deflated Dynamics Value Iteration

We present Deflated Dynamics Value Iteration (DDVI), after introducing its two key ingredients, matrix deflation and matrix splitting.

### 3.1 Matrix Deflation

Recall from the introduction that we would like to remove from $\mathcal{P}^\pi$ the subspace corresponding to the top $s$ eigen-structure with eigenvalues $\lambda_1, \ldots, \lambda_s$. We use the so-called deflation technique to displace, and in fact remove, some of the eigenvalues of the matrix $\mathcal{P}^\pi$ without changing the rest. This is done by subtracting a matrix $E$ of a certain form from $\mathcal{P}^\pi$ (see Section 4.2 of Saad 2011 for a discussion of deflation technique).

Let $\mathcal{P}^\pi$ be an $n \times n$ matrix with eigenvalues $\lambda_1, \ldots, \lambda_n$, sorted in decreasing order of magnitude with ties broken arbitrarily. Let $u^\top$ denote the conjugate transpose of $u \in \mathbb{C}^n$. We describe three ways of deflating this matrix and some properties of each approach as the following Facts.

**Fact 1** (Hotelling's deflation, Meirovitch 1980, Section 5.6). Assume $s \leq n$ linearly independent eigenvectors corresponding to $\{\lambda_i\}_{i=1}^s$ exist. Write $\{u_i\}_{i=1}^s$ and $\{v_i\}_{i=1}^s$ to denote the top $s$ right and left eigenvectors scaled to satisfy $u_i^\top v_i = 1$ and $u_i^\top v_j = 0$ for all $1 \leq i \neq j \leq s$. If $E_s = \sum_{i=1}^s \lambda_i u_i v_i^\top$, then $\rho(\mathcal{P}^\pi - E_s) = |\lambda_{s+1}|$.

Hotelling deflation makes $\rho(\mathcal{P}^\pi - E_s)$ small by eliminating the top $s$ eigenvalues of $\mathcal{P}^\pi$, but requires both right and left eigenvectors of $\mathcal{P}^\pi$. Wielandt's deflation, in contrast, requires only right eigenvectors.

**Fact 2** (Wielandt's deflation, Soto & Rojo 2006, Theorem 5). Assume $s \leq n$ linearly independent eigenvectors corresponding to $\{\lambda_i\}_{i=1}^s$ exist. Write $\{u_i\}_{i=1}^s$ to denote the top $s$ linearly independent right eigenvectors. Assume that vectors $\{v_i\}_{i=1}^s$, which are not necessarily the left eigenvectors, satisfy $u_i^\top v_i = 1$ and $u_i^\top v_j = 0$ for all $1 \leq i \neq j \leq s$. If $E_s = \sum_{i=1}^s \lambda_i u_i v_i^\top$, then $\rho(\mathcal{P}^\pi - E_s) = |\lambda_{s+1}|$.

Wielandt's deflation, however, still requires right eigenvectors, which are sometimes numerically unstable to compute. The Schur deflation, in contrast, only requires Schur vectors, which are stable to compute (Golub & Van Loan, 2013, Sections 7.3). For any $\mathcal{P}^\pi \in \mathbb{R}^{n \times n}$, the Schur decomposition has the form $\mathcal{P}^\pi = URU^\top$, where $R$ is an upper triangular matrix with $R_{ii} = \lambda_i$ for $i = 1, \ldots, n$, and $U$ is a unitary matrix. We write $u_i$ to denote the $i$-th column of $U$ and call it the $i$-th Schur vector for $i = 1, \ldots, n$. Specifically, the QR iteration computes the top $s$ eigenvalues and Schur vectors.

**Fact 3** (Schur deflation, Saad 2011, Proposition 4.2). Let $s \leq n$. Write $\{u_i\}_{i=1}^s$ to denote the top $s$ Schur vectors. If $E_s = \sum_{i=1}^s \lambda_i u_i u_i^\top$, then $\rho(\mathcal{P}^\pi - E_s) = |\lambda_{s+1}|$.

If an $E_s$ satisfies the conditions of Facts 1, 2, or 3, we say it is a *rank-$s$ deflation matrix*. Later, it will be needed that for $E_s = \sum_{i=1}^s \lambda_i u_i v_i^\top$, the identity

$$(I - \alpha\gamma E_s)^{-1} = I + \sum_{i=1}^s \frac{\alpha\gamma\lambda_i}{1 - \alpha\gamma\lambda_i} u_i v_i^\top \quad (1)$$

should hold. Indeed, the conditions of Facts 1, 2, or 3 do imply equation 1.

## 3.2 Matrix Splitting: Successive over-relaxation

We now describe the second key ingredient of DDVI: matrix splitting. Recall that the policy evaluation problem is the problem of finding the value function $V^\pi$ such that $T^\pi V^\pi = V^\pi$. This is in fact a system of linear equations:

$$(I - \gamma \mathcal{P}^\pi)V^\pi = r^\pi. \tag{2}$$

DDVI's approach to solve this equation is based on the Successive over-relaxation (SOR) algorithm, an iterative approach to solve linear systems.

To introduce SOR in its general form, consider a matrix $A \in \mathbb{R}^{n \times n}$ and the linear system $Az = b$. SOR starts by splitting $A$ in the form of $A = B + C + D$. Let $\alpha \neq 0$. Then, $z$ solves the linear system $Az = b$ if and only if

$$(D + \alpha B)z = \alpha b - (\alpha C + (\alpha - 1)D)z,$$

which, in turn, holds if and only if

$$z = (D + \alpha B)^{-1}(\alpha b - (\alpha C + (\alpha - 1)D)z),$$

provided that $D + \alpha B$ is invertible. SOR attempts to find a solution through the fixed-point iteration

$$z^{k+1} = (D + \alpha B)^{-1}(\alpha b - (\alpha C + (\alpha - 1)D)z^k).$$

Note that classical SOR uses a lower triangular $B$, upper triangular $C$, and diagonal $D$. Here, we generalize the standard derivation of SOR to any splitting $A = B + C + D$.

In the case of the policy evaluation problem of equation 2, we have $A = I - \gamma \mathcal{P}^\pi$ and $b = r^\pi$. For any $E \in \mathbb{R}^{n \times n}$, we can consider the splitting

$$I - \gamma \mathcal{P}^\pi = -\gamma E - \gamma(\mathcal{P}^\pi - E) + I,$$

where we chose $B = -\gamma E$, $C = -\gamma(\mathcal{P}^\pi - E)$, and $D = I$. With these choices, the SOR iteration for PE is

$$V^{k+1} = (I - \alpha\gamma E)^{-1}(\alpha r^\pi + ((1-\alpha)I + \alpha\gamma(\mathcal{P}^\pi - E))V^k). \tag{3}$$

Notice that $V^\pi$ is the fixed point of this iterative procedure: $V^\pi = (I - \alpha\gamma E)^{-1}(\alpha r^\pi + ((1-\alpha)I + \alpha\gamma(\mathcal{P}^\pi - E))V^\pi)$. Whether these iterations converge to the fixed point, however, depends on the choice of $\alpha$ and $E$. As an example, for $\alpha = 1$ and $E = 0$, we recover the original VI, which is convergent with the convergence rate of $O(\gamma^k)$. Next, we propose particular choices that lead to not only convergence but acceleration.

## 3.3 Deflated Dynamics Value Iteration

We are now ready to introduce DDVI. Let $E_s = \sum_{i=1}^s \lambda_i u_i v_i^\top$ be a rank-$s$ deflation matrix of $\mathcal{P}^\pi$ satisfying one of the conditions of Facts 1, 2, or 3. The SOR iteration for PE with deflation matrix $E_s$ (cf. equation 3) is

$$V^{k+1} = (I - \alpha\gamma E_s)^{-1}(\alpha r^\pi + ((1-\alpha)I + \alpha\gamma(\mathcal{P}^\pi - E_s))V^k). \tag{4}$$

By benefitting from equation 1, we can write it as

$$W^{k+1} = (1-\alpha)V^k + \alpha r^\pi + \alpha\gamma\left(\mathcal{P}^\pi - \sum_{i=1}^s \lambda_i u_i v_i^\top\right)V^k$$

$$V^{k+1} = \left(I + \sum_{i=1}^s \frac{\alpha\gamma\lambda_i}{1 - \alpha\gamma\lambda_i} u_i v_i^\top\right)W^{k+1}. \tag{5}$$

We call this method *Deflated Dynamics Value Iteration* (DDVI). Theorem 3.1 and Corollary 3.2 describe the rate of convergence of DDVI for PE.

**Theorem 3.1.** *Let $\pi$ be a policy and let $\lambda_1, \ldots, \lambda_n$ be the eigenvalues of $\mathcal{P}^\pi$ sorted in decreasing order of magnitude with ties broken arbitrarily. Let $s \leq n$. Let $E_s = \sum_{i=1}^{s} \lambda_i u_i v_i^\top$ be a rank-s deflation matrix of $\mathcal{P}^\pi$ satisfying the conditions of Facts 1, 2, or 3. For $0 < \alpha \leq 1$, DDVI equation 5 exhibits the rate*[1]

$$\|V^k - V^\pi\| = \tilde{O}(|\lambda|^k \|V^0 - V^\pi\|)$$

*as $k \to \infty$, where*

$$\lambda = \max_{1 \leq i \leq s, s+1 \leq j \leq n} \left\{ \left| \frac{1-\alpha}{1-\alpha\gamma\lambda_i} \right|, |1-\alpha+\alpha\gamma\lambda_j| \right\}.$$

When $\alpha = 1$, we can simplify DDVI equation 5 as follows.

**Corollary 3.2.** *In the setting of Theorem 3.1, if $\alpha = 1$, then DDVI equation 5 simplifies to*

$$W^{k+1} = \gamma(\mathcal{P}^\pi - E_s)V^k + r^\pi,$$

$$V^{k+1} = \left( I + \sum_{i=1}^{s} \frac{\gamma\lambda_i}{1-\gamma\lambda_i} u_i v_i^\top \right) W^{k+1},$$

*and the the rate simplifies to $\|V^k - V^\pi\| = \tilde{O}(|\gamma\lambda_{s+1}|^k \|V^0 - V^\pi\|)$ as $k \to \infty$.*

Let us compare this convergence rate with the original VI's. The rate for VI is $O(\gamma^k)$, which is slow when $\gamma \approx 1$. By choosing $E_s$ to deflate the top $s$ eigenvalues of $\mathcal{P}^\pi$, DDVI has the rate of $O(|\gamma\lambda_{s+1}|^k)$, which is exponentially faster whenever $\lambda_{s+1}$ is smaller than 1. The exact behaviour depends on the spectrum of the Markov chain (whether eigenvalues are close to 1 or far from them) and the number of eigenvalues we decided to deflate by $E_s$.

In Section 4, we discuss a practical implementation of DDVI based on the power iteration and QR iteration and implement it in the experiments of Section 6. We find that smaller values of $\alpha$ lead to more stable iterations when using approximate deflation matrices. We also show how we can use DDVI in the RL setting in Section 5.

This version of DDVI equation 5 applies to the policy evaluation (PE) setup. The challenge in extending DDVI to the Control setup is that $\mathcal{P}^\pi$ changes throughout the iterations, and so should the deflation matrix $E_s$. However, when $s = 1$, the deflation matrix $E_1$ can be kept constant, and we utilize this fact in Section 4.1.

## 4 Computing Deflation Matrix $E$

The application of DDVI requires practical means of computing the deflation matrix $E_s$. In this section, we provide three approaches as examples.

### 4.1 Rank-1 DDVI for PE and Control

Recall that for any stochastic $n \times n$ matrix, the vector $\mathbf{1} = [1, \ldots, 1]^\top \in \mathbb{R}^n$ is a right eigenvector corresponding to eigenvalue 1. This allows us to easily obtain a rank-1 deflation matrix $E_1$ for $\mathcal{P}^\pi$ for any policy $\pi$.

Let $E_1 = \mathbf{1}v^\top$ with $v \in \mathbb{R}^n$ be a vector with non-negative entries satisfying $v^\top \mathbf{1} = 1$. This rank-1 Wielandt's deflation matrix can be used for DDVI (PE) as in Section 2, but we can also use it for the Control version of DDVI.

We define the rank-1 DDVI for Control as

$$W^{k+1} = \max_\pi \{r^\pi + \gamma(\mathcal{P}^\pi - E_1)W^k\} = \max_\pi \{r^\pi + \gamma\mathcal{P}^\pi W^k\} - \gamma(v^\top W^k)\mathbf{1}. \tag{6}$$

Here, we benefitted from the fact that $E_1$ is not a function of $\pi$ in order to take it out of the $\max_\pi$. The value function $V$ can be computed as

$$V^k = \left( I + \frac{\gamma}{1-\gamma}\mathbf{1}v^\top \right) W^k. \tag{7}$$

---

[1]The precise meaning of the $\tilde{O}$ notation is $\|V^k - V^\pi\| = O(|\lambda + \epsilon|^k \|V^0 - V^\pi\|)$ as $k \to \infty$ for any $\epsilon > 0$.

Note that the iteration over $W^k$ does not depend on $V^k$, so we do not need to compute $V$ at each iteration – only when we need it. Compared to equation 5 of DDVI (PE), we set $\alpha = 1$ for simplicity. We have the following guarantee for rank-1 DDVI for Control.

**Theorem 4.1.** *The DDVI for Control algorithm (equation 6 and equation 7) exhibits the rate*

$$\|V^k - V^\star\| \le \frac{2}{1-\gamma}\gamma^k\|V^0 - V^\star\|,$$

*for $k = 0, 1, \ldots$. Furthermore, if there exist a unique optimal policy $\pi^\star$, then*

$$\|V^k - V^\star\| = \tilde{O}(|\gamma\lambda_2|^k\|V^0 - V^\star\|)$$

*as $k \to \infty$, where $\lambda_2$ is the second eigenvalue of $\mathcal{P}^{\pi^\star}$ (The $\tilde{O}$ notation is as defined in Theorem 3.1).*

**Discussion.** Although rank-1 DDVI for Control does accelerate the convergence $V^k \to V^\star$, a subtle point to note is that the greedy policy $\pi^k$ obtained from $V^k$ is not affected by the rank-1 deflation. Briefly speaking, this is because adding a constant to $V^k$ through $\mathbf{1}$ has no effect on the $\arg\max$ computation used in the greedy policy. Indeed, the maximizer $\pi^{k+1}$ in equation 6 is the same as $\arg\max_\pi\{r^\pi + \gamma\mathcal{P}^\pi V^k\}$, produced by the (non-deflated) value iteration, when $W^0 = V^0$. Having the term $v^\top\mathbf{1}W^k$ in the update $W^{k+1} = r^\pi + \gamma(\mathcal{P}^{\pi^{k+1}} - v^\top\mathbf{1})W^k$ adds the same constant to all states, so it does not change the maximizer of the next policy.

## 4.2 DDVI with Automatic Power Iteration

According to Fact 2, we can construct the deflation matrix $E_s$ for $\mathcal{P}^\pi$ if we have its top $s$ right eigenvectors. The top eigenvector $u_1 = \mathbf{1}$ is known, which as we saw in Section 4.1, gives $E_1$ and rank-1 DDVI. We show that we can start with $s = 1$ and run rank-1 DDVI, and using the calculations already done in DDVI updates, estimate the next top eigenvectors and increase the deflation rank.

Assume that the top $s + 1$ eigenvalues of $\mathcal{P}^\pi$ have distinct magnitude, i.e., $1 = \lambda_1 > |\lambda_2| > \cdots > |\lambda_{s+1}|$. Let $E_s = \sum_{i=1}^s \lambda_i u_i v_i^\top$ be a rank-$s$ deflation matrix of $\mathcal{P}^\pi$, as in Fact 2. Consider the DDVI algorithm in equation 5 with $\alpha = 1$ as in Corollary 3.2. If $V^0 = 0$, since

$$(\mathcal{P}^\pi - E_s)(I - \gamma E_s)^{-1} = \mathcal{P}^\pi - E_s$$

(verified as equation 10 in Appendix B), we have

$$W^k = \sum_{i=0}^{k-1} \gamma^i (\mathcal{P}^\pi - E_s)^i r^\pi.$$

Therefore, $\frac{1}{\gamma^k}(W^{k+1} - W^k) = (\mathcal{P}^\pi - E_s)^k r^\pi$ is the iterates of a power iteration for the matrix $(\mathcal{P}^\pi - E_s)$ starting from initial vector $r^\pi$. As discussed in Fact 2, the top eigenvalue of $(\mathcal{P}^\pi - E_s)$ is $\lambda_{s+1}$. For large $k$, we expect $\frac{W^{k+1} - W^k}{\|W^{k+1} - W^k\|} \approx w$, where $w$ is the top eigenvector of $(\mathcal{P}^\pi - E_s)$. With $w$, we can recover the $(s+1)$-th right eigenvector of $\mathcal{P}^\pi$ through the formula (Bru et al., 2012, Proposition 5)

$$u_{s+1} = w - \sum_{i=1}^s \frac{\lambda_i v_i^\top w}{\lambda_i - \lambda_{s+1}} u_i.$$

Leveraging this observation, *DDVI with Automatic Power Iteration* (AutoPI) computes an approximate rank-$s$ deflation matrix $E_s$ while performing DDVI: Start with a rank-1 deflation matrix, using the first right eigenvector $u_1 = \mathbf{1}$, and carry out DDVI iterations. If a certain error criteria is satisfied, use $W^{k+1} - W^k$ to approximate the second right eigenvector $u_2$. Then, update the deflation matrix to be rank 2, and gradually increase the deflation rank. We formalize this approach in Algorithm 1.

---

**Algorithm 1** DDVI with AutoPI

---

1: Initialize $C, \epsilon$                                                               ▷ For example, $C = 10$, $\epsilon = 10^{-4}$
2: **function** DDVI($(s, V, K, \{\lambda_i\}_{i=1}^s, \{u_i\}_{i=1}^s)$)
3:     $V^0 = V$, $c = 0$, $E_s = \sum_{i=1}^s \lambda_i u_i v_i^\top$ as in Fact 2
4:     **for** $k = 0, \ldots, K-1$ **do**
5:         **if** $c \geq C$ and $\left| \frac{w^{k+1}}{\|w^{k+1}\|_2} - \frac{w^k}{\|w^k\|_2} \right| < \epsilon$ **then**
6:             $\lambda_{s+1} = (w^k)^\top w^{k+1} / \left(\gamma \|w^k\|_2^2\right)$, $u_{s+1} = w^{k+1} - \sum_{i=1}^s \frac{\lambda_i v_i^\top w^{k+1}}{\lambda_i - \lambda_{s+1}} u_i$
7:             Return DDVI($s{+}1, V^k, K{-}c, \{\lambda_i\}_{i=1}^{s+1}, \{u_i\}_{i=1}^{s+1}$)
8:         **else**
9:             $W^{k+1} = \gamma(\mathcal{P}^\pi - E_s)V^k + r^\pi$, $V^{k+1} = (I - \gamma E_s)^{-1} W^{k+1}$
10:            $w^{k+1} = W^{k+1} - W^k$, $w^k = W^k - W^{k-1}$
11:            $c = c + 1$
12:     Return $V^K$
13: Initialize $V^0$ and $u_1 = \mathbf{1}$, $\lambda_1 = 1$
14: DDVI($1, V^0, K, \lambda_1, u_1$)

---

### 4.3 Rank-$s$ DDVI with the QR Iteration.

Recall that the QR iteration approximates top $s$ Schur vectors. Algorithm 2 uses the QR Iteration to construct the rank-$s$ Schur deflation matrix and performs DDVI. Compared to the standard VI, rank-$s$ DDVI requires additional computation for the QR iteration.

---

**Algorithm 2** Rank-$s$ DDVI with QR Iteration

---

Initialize $\alpha$, $s$, and $V^0$
$\{\lambda_i\}_{i=1}^s, \{u_s\}_{i=1}^s = \text{QRiteration}(\mathcal{P}^\pi, s)$
**for** $k = 0, \ldots, K-1$ **do**
    $W^{k+1} = \alpha\gamma(\mathcal{P}^\pi - \sum_{i=1}^s \lambda_i u_i u_i^\top)V^k + (1-\alpha)V^k + \alpha r^\pi$
    $V^{k+1} = \left(I + \sum_{i=1}^s \frac{\alpha\gamma\lambda_i}{1-\alpha\gamma\lambda_i} u_i u_i^\top\right) W^{k+1}$

---

The QR iteration of Algorithm 2 can be carried out in an automated manner similar to AutoPI. We refer to this as AutoQR and formally describe the algorithm in Appendix C.

## 5 Deflated Dynamics Temporal Difference Learning

Many practical RL algorithms such as Temporal Difference (TD) Learning (Sutton, 1988; Tsitsiklis & V. Roy, 1997), Q-Learning (Watkins, 1989), Fitted Value Iteration (Gordon, 1995), and DQN (Mnih et al., 2015) can be viewed as sample-based variants of VI. Consequently, the slow convergence in the case of $\gamma \approx 1$ has also been observed for these algorithms by Szepesvári (1997); Even-Dar & Mansour (2003); Wainwright (2019) for TD Learning and by Munos & Szepesvári (2008); Farahmand et al. (2010); Chen & Jiang (2019); Fan et al. (2020) for Fitted VI and DQN. Here we introduce *Deflated Dynamics Temporal Difference Learning* (DDTD) as a sample-based variant of DDVI. To start, recall that the tabular temporal difference (TD) learning performs the updates

$$V^{k+1}(X_k) = V^k(X_k) + \eta_k(X_k)[r^\pi(X_k) + \gamma V^k(X_k') - V^k(X_k)],$$

where $\eta_k(X_k)$ is a state-dependent stepsize, $(X_k, X_k')$ are random samples from the environment such that $X_k'$ is the subsequent state following $X_k$, and $r^\pi(X_k)$ is the expected immediate reward obtained by following policy $\pi$ from state $X_k$. In practice, we may choose a state-independent $\eta_k(X_k) = \eta_k$, which would still result in convergence to the true value function if each state is visited often enough.

---

**Algorithm 3** Rank-$s$ DDTD with QR iteration

---

1: Initialize $\alpha$, $s$, $W$, $V$, $E_s$, $\hat{\mathcal{P}}^\pi$, and $K$
2: **for** $k = 1, 2, \ldots$ **do**
3:     Choose $X_k$ uniformly at random.
4:     Sample $(\pi(X_k), R_k, X_k')$ from environment
5:     Update $\hat{\mathcal{P}}$ with $(X_k, \pi(X_k), R_k, X_k')$
6:     **if** $k \bmod K = 0$ **then**
7:         $\{\lambda_i\}_{i=1}^s, \{u_s\}_{i=1}^s = \text{QRiteration}(\hat{\mathcal{P}}^\pi, \text{s})$
8:         Update $E_s \leftarrow \sum \lambda_i u_i u_i^\top$
9:         $W \leftarrow (I - \alpha\gamma E_s)V$
10:    $W(X_k) \leftarrow W(X_k) + \eta_k(X_k)\big[\alpha R_k + \alpha\gamma V(X_k') - \alpha\gamma(E_s V)(X_k) + (1-\alpha)V(X_k) - W(X_k)\big]$
11:    $V \leftarrow \left(I + \sum_{i=1}^s \frac{\alpha\gamma\lambda_i}{1-\alpha\gamma\lambda_i} u_i u_i^\top\right) W$

---

TD is a sample-based variant of VI for PE. Two key ingredients of TD learning are the random coordinate updates and the *temporal difference* error $r^\pi(X) + \gamma V(X') - V(X)$, whose conditional expectation is equal to $(T^\pi V - V)(X)$. Applying the same ingredients to DDVI, we obtain DDTD. The improved convergence rate for DDVI compared to VI suggests that DDTD may also exhibit improved convergence rates.

Specifically, we define DDTD as

$$W^{k+1}(X_k) = W^k(X_k) + \eta_k(X_k)\big[(1-\alpha)V^k(X_k) + \alpha\left(r^\pi(X_k) + \gamma V(X_k') - \gamma(E_s V^k)(X_k)\right) - W^k(X_k)\big]$$

$$V^{k+1} = \left(I + \sum_{i=1}^s \frac{\alpha\gamma\lambda_i}{1-\alpha\gamma\lambda_i} u_i v_i^\top\right)W^{k+1}, \tag{8}$$

where $E_s = \sum_{i=1}^s \lambda_i u_i v_i^\top$ is a rank-$s$ deflation matrix of $\mathcal{P}^\pi$ and $\{X_k, X_k'\}_{k=0,1,\ldots}$ are i.i.d. random variables such that $X_k' \sim \mathcal{P}^\pi(\cdot \mid X_k)$ and $X_k \sim \text{Unif}(\mathcal{X})$. The $W^{k+1}$-update notation means $W^{k+1}(x) = W^k(x)$ for all $x \neq X_k$.

DDTD is the sample-based variant of DDVI performing asynchronous updates. The following result provides almost sure convergence of DDTD.

**Theorem 5.1.** *Let $\eta_k(x) = (\sum_{i=0}^k \mathbf{1}_{X_i=x})^{-1}$ when $X_k = x$. For $\alpha = 1$, DDTD equation 8 converges to $V^\pi$ almost surely.*

The following result describes the asymptotic convergence rate of DDTD.

**Theorem 5.2.** *Let $\lambda_1, \ldots, \lambda_n$ be the eigenvalues of $\mathcal{P}^\pi$ sorted in decreasing order of magnitude with ties broken arbitrarily. Let $\eta_k(x) = C/(k+1)$ with the constant $C > \frac{n}{2\lambda_{DDTD}}$, where $\lambda_{DDTD} = \min_{\lambda \in \{\lambda_{s+1}, \ldots, \lambda_n\}} Re(1-\gamma\lambda)$. For $\alpha = 1$, DDTD equation 8 exhibits the rate $\mathbf{E}[\|V^k - V^\pi\|^2] = O(k^{-1})$.*

We note that in Theorem 5.2, as the rank of DDTD increases, $\lambda_{\text{DDTD}}$ also increases, and this implies that higher rank DDTD has a larger range of convergent step size compared to the plain TD learning.

### 5.1 Implementation of DDTD

To implement DDTD practically, we take a hybrid of model-free and model-based approach for obtaining $E_s$. At each iteration, the agent uses the new samples to update an approximate model $\hat{\mathcal{P}}^\pi$ of the true dynamics. This updated approximate model is used to compute $E_s$. We update $E_s$ every $K$-th iterations since the change in $\hat{\mathcal{P}}^\pi$ and consequently the resulting $E_s$ at each iteration is small. Whenever a new $E_s$ is computed, we set $W$ to be $(I - \alpha\gamma E_s)V$. This step ensures that $V$ smoothly converges to $V^\pi$ by updating all coordinates of $W$ at once based on new $E_s$. Also, we use the random sample reward $R$ in place of the expected reward $r^\pi(X_k)$. We formally state the algorithm as Algorithm 3.

We note that DDTD simultaneously uses samples to learn the model and to update the value function, making the DDTD framework more effective than model-free learning and more computationally efficient than a purely model-based approach.

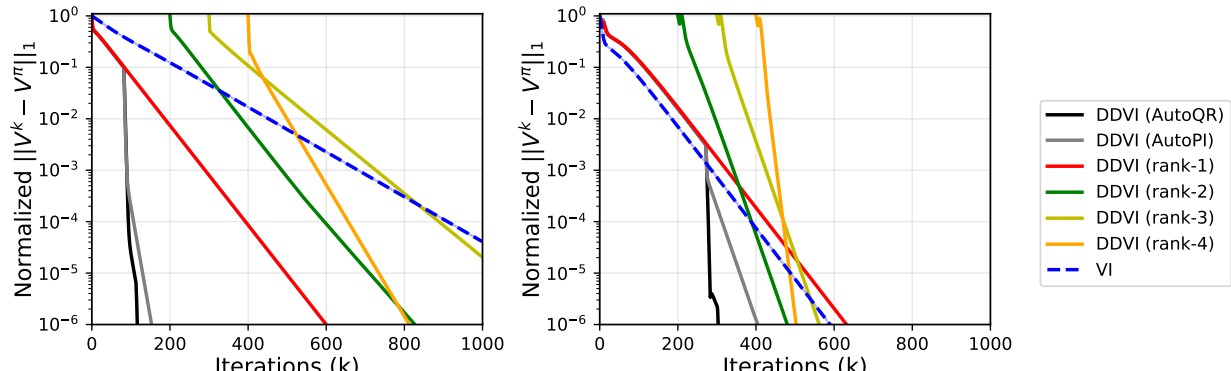

Figure 1: Comparison of DDVI with different ranks, AutoPI, and AutoQR against VI in (left) Maze and (right) Chain Walk. The plots for DDVI do not start at iteration 0 because the costs of computing $E_s$ through QR iterations are incorporated as rightward shifts. Rate of DDVI with AutoPI and AutoQR changes when $E_s$ is updated.

## 6    Experiments

For our experiments, we use the following environments: Maze with $5 \times 5$ states and 4 actions, Cliffwalk with $3 \times 7$ states with 4 actions, Chain Walk with 50 states with 2 actions, and random Garnet MDPs (Bhatnagar et al., 2009) with 200 states. The discount factor is set to $\gamma = 0.99$ for the comparison of DDVI with different ranks and the DDTD experiments, and $\gamma = 0.995$ in other experiments. Appendix D provides full definitions of the environments and policies used for PE. All experiments were carried out on local CPUs. We report the normalized error of $V^k$ defined as $\|V^k - V^\pi\|_1/\|V^\pi\|_1$.[2]

**DDVI with AutoPI, AutoQR, different fixed ranks.**    In Figure 1, we compare rank-$s$ DDVI for multiple values of $s$ and DDVI with AutoPI and AutoQR against the VI in Chain Walk and Maze. In our plots, we incorporate the cost of the QR iteration for a fair comparison. We consider the cost of a QR iteration to be $msC$ where $m$ is the number of QR iterations we applied, $s$ is the rank of DDVI, and $C$ is the cost of VI per iteration, and plot rank-$s$ DDVI starting from $ms$ iterations in Figures 1. In this experiment, QR iteration (Section 2) is used 100 times to calculate $E_s$ with Schur vectors. In almost all cases, DDVI exhibits a significantly faster convergence rate compared to VI. Aligned with the theory, we observe that higher ranks achieve better convergence rates. Also, as DDVI with AutoPI and AutoQR progress and update $E_s$, their convergence rate improves.

**DDVI and other baselines.**    We perform an extensive comparison of DDVI against the prior accelerated VI methods: Safe Accelerated VI (S-AVI)(Goyal & Grand-Clément, 2022), Anderson VI (Geist & Scherrer, 2018), PID VI (Farahmand & Ghavamzadeh, 2021), and Anchored VI (Lee & Ryu, 2023). In this experiment, we use the Implicitly Restarted Arnoldi Method (Lehoucq et al., 1998) from SciPy package to calculate the eigenvalues and eigenvectors for DDVI. Figure 2 (top-left) shows the convergence behaviour of the algorithms by iteration count in 20 randomly generated Garnet environments, where our algorithms outperform all the baselines. This comparison might not be fair as the amount of computation needed for each iteration of algorithms is not the same. Therefore, in Figure 2 (top-right), we compare the algorithms by wall clock time. It can be seen that rank-2 DDVI initially has to spend time to calculate $E_s$ before it starts to update the value function, but after the slow start, it has the fastest rate. The fast rate can compensate for the initial time if a high accuracy is needed. DDVI with AutoQR and rank-1 DDVI show a fast convergence from the beginning.

We further investigate how the algorithms scale with the size of MDP and the discount factor. Figure 2 (bottom-left) shows the runtime to reach a normalized error of $10^{-4}$ as a function of the number of states.

---

[2]The source code for the experiments can be found at https://github.com/adaptive-agents-lab/ddvi.

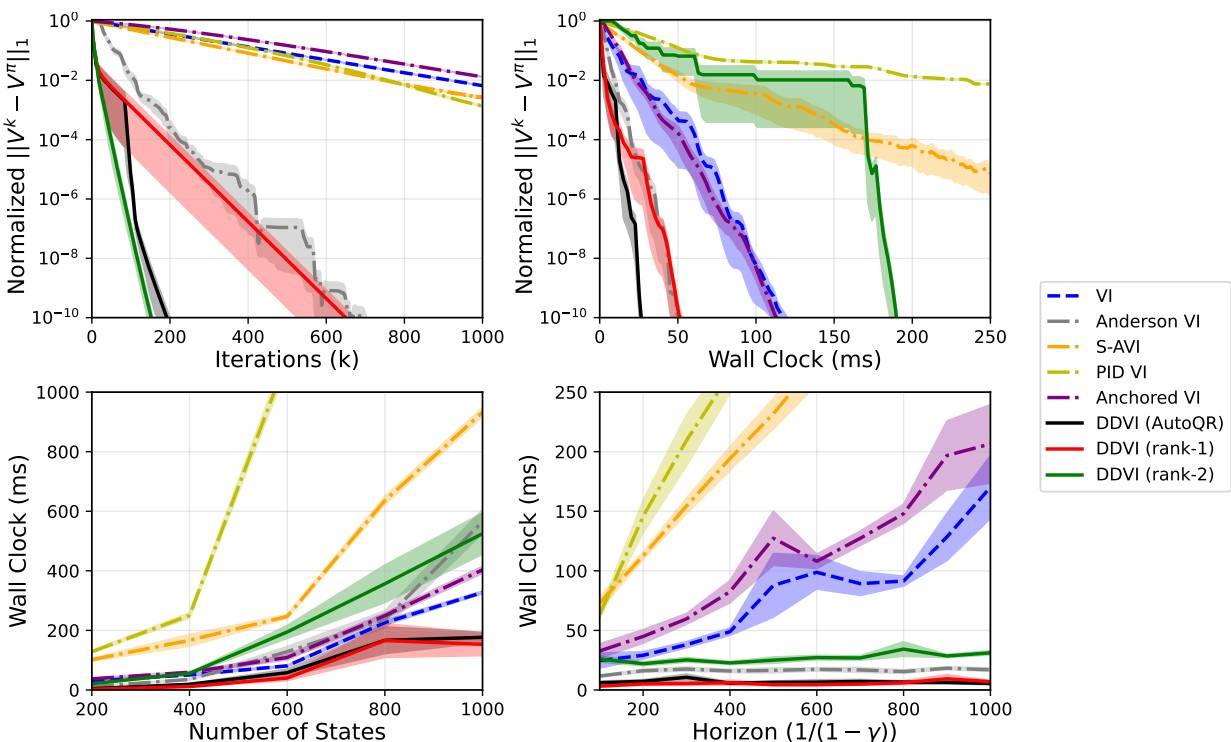

Figure 2: Comparison of DDVI with other accelerated VIs. Normalized errors are shown against the iteration number (top-left) and wall clock time (top-right). Runtimes to reach normalized error of $10^{-4}$ is shown against the number of states (bottom-left) and horizon $1/(1-\gamma)$ (bottom-right). Plots are average of 20 randomly generated Garnet MDPs with shaded areas showing the standard error.

DDVI with AutoQR and rank-1 DDVI show the best scaling. Note that even rank-2, for which the calculation of $E_s$ is non-trivial, also scales competitively. Figure 2 (bottom-right) show the scaling behaviour with horizon of the MDP, which is $1/(1-\gamma)$. We measure the runtime to reach a normalized error of $10^{-4}$ for the horizon ranging from 100 to 1000 which corresponds to $\gamma$ ranging from 0.99 to 0.999. Remarkably, DDVI algorithms have a low constant runtime even with long horizon tasks with $\gamma \approx 1$. This is aligned with our theoretical result, as the rate $\gamma|\lambda_{s+1}|$ remains small as $\gamma$ approaches 1.

**Rank-$s$ DDTD with QR iteration.** We compare DDTD with TD Learning and Dyna (Sutton, 1990; Peng & Williams, 1993) which simultaneously use samples to learn the model and to update the value function. For the sake of making the setting more realistic, we consider the case where the approximate model $\hat{\mathcal{P}}$ cannot exactly learn the true dynamics. Figure 3 shows that DDTD with large enough rank can outperform TD. Note that unlike Dyna, DDTD does not suffer from model error, which shows that DDTD only uses the model for acceleration and is not a pure model-based algorithm.

## 7 Conclusion

In this work, we proposed a framework for accelerating VI through matrix deflation and matrix splitting. We theoretically analyzed the proposed methods, DDVI and DDTD, and presented experimental results showing speedups in various setups. The positive experimental results demonstrate that matrix deflation to be a promising technique that may be applicable to a broader range of RL algorithms.

One direction of future work is to extend the theoretical analysis DDVI for Control using a general rank-$s$ deflation matrix. Theoretically analyzing other RL methods combined with matrix deflation is another

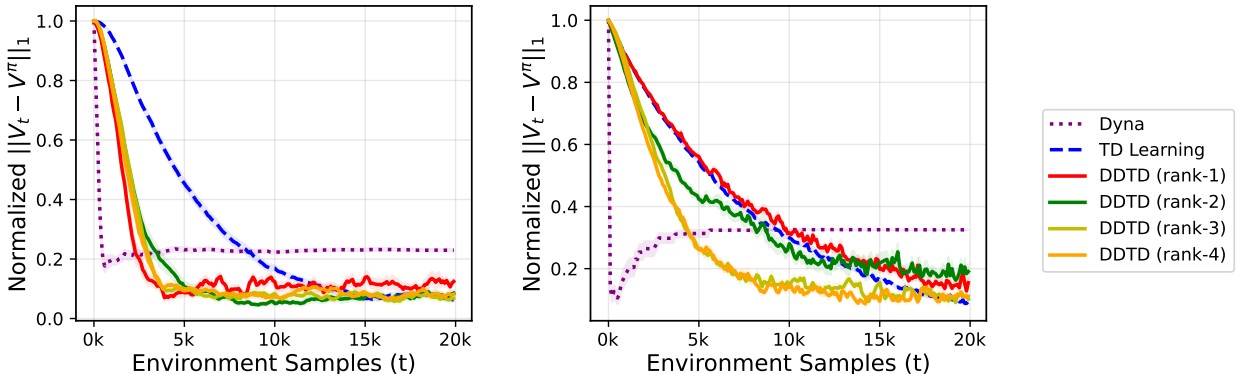

Figure 3: Comparison of DDTD with QR iteration, Dyna, and TD learning in (left) Chain Walk (right) Maze.

interesting direction. Another direction is incorporating the DDVI framework with a function approximator, such as a deep neural network, to handle large-scale or continuous state and action spaces.

### Acknowledgments

We thank the anonymous reviewers, the action editor, as well as the members of the Adaptive Agents (Adage) Lab, in particular Tyler Kastner, for their constructive feedback. JL and EKR were supported by the National Research Foundation of Korea (NRF) grant funded by the Korean government (No.RS-2024-00421203). AMF acknowledges the funding from the Natural Sciences and Engineering Research Council of Canada (NSERC) through the Discovery Grant program (2021-03701). Resources used in preparing this research were provided, in part, by the Province of Ontario, the Government of Canada through CIFAR, and companies sponsoring the Vector Institute.

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

# A   Prior Works

**Acceleration in RL.**   Prioritized sweeping and its several variants (Moore & Atkeson, 1993; Peng & Williams, 1993; McMahan & Gordon, 2005; Wingate et al., 2005; Andre et al., 1997; Dai et al., 2011) specify the order of asynchronous value function updates, which may lead to accelerated convergence to the true value function. Speedy Q-learning (Azar et al., 2011) changes the update rule of Q-learning and employs aggressive learning rates to accelerate the convergence. Sidford et al. (2023) accelerate the computation of $\mathcal{P}^\pi V^k$ by sampling it with variance reduction techniques.

Recently, there has been a growing body of research that explores the application of acceleration techniques of other areas of applied math to planning and RL: Geist & Scherrer (2018); Sun et al. (2021); Ermis & Yang (2020); Park et al. (2022); Ermis et al. (2021); Shi et al. (2019) apply Anderson acceleration of fixed-point iterations, Lee & Ryu (2023) apply Anchor acceleration of minimax optimization, Vieillard et al. (2020); Goyal & Grand-Clément (2022); Grand-Clément (2021); Bowen et al. (2021); Akian et al. (2022) apply Nesterov acceleration of convex optimization, and Farahmand & Ghavamzadeh (2021) apply ideas inspired by PID controllers in control theory.

Specifically, regarding the variants of VI used in our experiments, S-AVI (Goyal & Grand-Clément, 2022) exhibits $O(\gamma_s^k)$-rate where $\gamma_s$ satisfies $\gamma \le \gamma_s \le 1$, PID VI (Farahmand & Ghavamzadeh, 2021) exhibits a guaranteed $O\left(\left(\frac{\sqrt{1+\gamma}-\sqrt{1-\gamma}}{\sqrt{1+\gamma}+\sqrt{1-\gamma}}\right)^k\right)$ rate under reversible MDP (and possibly faster, depending on the MDP), and Anchored VI (Lee & Ryu, 2023) exhibits $O\left(\frac{(\gamma^{-1}-\gamma)(1+2\gamma-\gamma^{k+1})}{(\gamma^{k+1})^{-1}-\gamma^{k+1}}\right)$-rate in terms of Bellman error.

**Matrix deflation.**   Deflation techniques were first developed for eigenvalue computation (Meirovitch, 1980; Saad, 2011; Golub & Van Loan, 2013). Matrix deflation, eliminating top eigenvalues of a given matrix with leaving the rest of the eigenvalues untouched, has been applied to various algorithms such as the conjugate gradient algorithm (Saad et al., 2000), principal component analysis (Mackey, 2008), the generalized minimal residual method (Morgan, 1995), and nonlinear fixed-point iteration (Shroff & Keller, 1993) to improve the convergence.

Some prior work in RL has explored subtracting constants from the iterates of VI, resulting in methods that resemble our rank-1 DDVI. For discounted MDP, Devraj & Meyn (2021) propose relative Q-learning, which can roughly be seen as the Q-learning version of rank-1 DDVI for Control, and Bertsekas (1995) proposes an extrapolation method for VI, which can be seen as rank-1 DDVI with Schur deflation matrix for PE. White (1963) introduced relative value iteration in the context of undiscounted MDP with average reward, and several variants were proposed (Gupta et al., 2015; Sharma et al., 2020).

**Matrix splitting of value iteration.**   Matrix splitting has been studied in the RL literature to obtain an acceleration. Hastings (1968) and Kushner & Kleinman (1968) first suggested Gausss-Seidel iteration for computing value function. Kushner & Kleinman (1971) and Reetz (1973) applied Successive Over-Relaxation to VI and generalized Jacobi and Gauss-Seidel iterations. Porteus (1975) proposed several transformations of MDP that can be seen as a Gauss–Seidel variant of VI. Rakhsha et al. (2022) used matrix splitting with the approximate dynamics. Bacon & Precup (2016); Bacon (2018) analyzed planning with options through the matrix splitting perspective.

**Convergence analysis of TD learning**  Jaakkola et al. (1993) first proved the convergence of TD learning using the stochastic approximation (SA) technique. Borkar & Meyn (2000); Borkar (1998) suggested ODE-based framework to provide asymptotic convergence of SA including TD learning with an asynchronous update. Lakshminarayanan & Szepesvari (2018) study linear SA under i.i.d. noise with respect to mean square error, and Chen et al. (2020) study asymptotic convergence rate of linear SA under Markovian noise. Finite-time analysis of TD learning was first provided by Dalal et al. (2018) and extended to Markovian noise setting by Bhandari et al. (2018). Leveraging Lyapunov theory, Chen et al. (2023) establishes finite-time analysis of Markovian SA with an explicit bound.

## B  Proof of Theoretical Results

We prove the theoretical results in this appendix.

### B.1  Proof of Theorem 3.1

By the definition of DDVI (equation 4) and the property of fixed point $V^\pi$, which satisfies $V^\pi = (I - \alpha\gamma E)^{-1}(\alpha r^\pi + ((1-\alpha)I + \alpha\gamma(\mathcal{P}^\pi - E))V^\pi)$ as mentioned in Section 3.2, we have

$V^k - V^\pi =$

$(I - \alpha\gamma E_s)^{-1}\left[(1-\alpha)I + \alpha\gamma(\mathcal{P}^\pi - E_s)\right](V^{k-1} - V^\pi) =$

$(I - \alpha\gamma E_s)^{-1}\left[(1-\alpha)(I - \alpha\gamma E_s)^{-1} + \alpha\gamma(\mathcal{P}^\pi - E_s)(I - \alpha\gamma E_s)^{-1}\right]((1-\alpha)I + \alpha\gamma(\mathcal{P}^\pi - E_s))(V^{k-2} - V^\pi) =$

$(I - \alpha\gamma E_s)^{-1}\left[(1-\alpha)(I - \alpha\gamma E_s)^{-1} + \alpha\gamma(\mathcal{P}^\pi - E_s)(I - \alpha\gamma E_s)^{-1}\right]^{k-1}((1-\alpha)I + \alpha\gamma(\mathcal{P}^\pi - E_s))(V^0 - V^\pi),$

where $E_s$ is a rank-$s$ deflation matrix of $\mathcal{P}^\pi$ satisfying the conditions of Facts 1, 2, or 3. This implies that

$$\|V^k - V^\pi\| \le C \left\|((1-\alpha)(I - \alpha\gamma E)^{-1} + \alpha\gamma(\mathcal{P}^\pi - E)(I - \alpha\gamma E)^{-1})^{k-1}\right\| \|V^0 - V^\pi\|, \tag{9}$$

for some constant $C \in \mathbb{R}$.

First, suppose $E_s$ satisfies Facts 1 or 2. Let $U_s = [u_1, \ldots, u_s]$ and $V_s = [v_1, \ldots, v_s]$. Define $D_{s,f(\lambda_i)} = \mathrm{diag}(f(\lambda_1), \ldots, f(\lambda_s))$ for some function $f$. Then $U_s, V_s, D_{s,f(\lambda_i)}$ are $s \times s$ matrices and $E_s = U_s D_{s,\lambda_i} V_s^\top$. By Jordan decomposition, $\mathcal{P}^\pi = UJU^{-1}$ where $J$ is Jordan matrix with $J_{ii} = \lambda_i$ for $i = 1, \ldots, n$. Let $J_s$ be $s \times s$ submatrix of $J$ satisfying $(J_s)_{ij} = J_{ij}$ for $1 \le i, j \le s$. Then, by condition of Theorem 3.1, $J_s = \mathrm{diag}(\lambda_1, \ldots, \lambda_s) = D_{s,\lambda_i}$, and the $i$-th column of $U$ is $u_i$ for $1 \le i \le s$. By simple calculation, we get $\mathcal{P}^\pi U_s = U_s J_s = U_s D_{s,\lambda_i}$ and $(I - \alpha\gamma E_s)^{-1} = I + U_s D_{s,\frac{\gamma\alpha\lambda_i}{1-\gamma\alpha\lambda_i}} V_s^\top$.

Since

$$(\mathcal{P}^\pi - U_s D_{s,\lambda_i} V_s^\top)(U_s D_{s,\frac{\gamma\alpha\lambda_i}{1-\gamma\alpha\lambda_i}} V_s^\top) = U_s D_{s,\lambda_i} D_{s,\frac{\gamma\alpha\lambda_i}{1-\gamma\alpha\lambda_i}} V_s^\top - U_s D_{s,\lambda_i} D_{s,\frac{\gamma\alpha\lambda_i}{1-\gamma\alpha\lambda_i}} V_s^\top$$
$$= 0,$$

we get

$$(\mathcal{P}^\pi - E_s)(I - \alpha\gamma E_s)^{-1} = \mathcal{P}^\pi - E_s. \tag{10}$$

Then, the term on the right-hand side (RHS) of equation 9 can be written as

$$(1-\alpha)(I - \alpha\gamma E_s)^{-1} + \alpha\gamma(\mathcal{P}^\pi - E_s)(I - \alpha\gamma E_s)^{-1} =$$
$$(1-\alpha)\left(I + U_s D_{s,\frac{\gamma\alpha\lambda_i}{1-\gamma\alpha\lambda_i}} V_s^\top\right) + \alpha\gamma\left(UJU^{-1} - U_s D_{s,\lambda_i} V_s^\top\right) =$$
$$(1-\alpha)I + \alpha\gamma UJU^{-1} + U_s D_{s,\frac{(\lambda_i\gamma-1)\gamma\alpha^2\lambda_i}{1-\gamma\alpha\lambda_i}} V_s^\top =$$
$$U\left((1-\alpha)I + \alpha\gamma J + e_{1:s} D_{s,\frac{(\lambda_i\gamma-1)\gamma\alpha^2\lambda_i}{1-\gamma\alpha\lambda_i}} V_s^\top U\right) U^{-1},$$

where $e_i \in \mathbb{R}^n$ is the $i$-th unit vector and $e_{1:s} = [e_1, \ldots, e_s]$, and $(1-\alpha)I + \alpha\gamma J + e_{1:s}D_{s,\frac{(\lambda_i\gamma-1)\gamma\alpha^2\lambda_i}{1-\gamma\alpha\lambda_i}}V_s^\top U$ is an upper triangular matrix with diagonal entries $\frac{(1-\alpha)}{1-\alpha\gamma\lambda_1}, \ldots, \frac{(1-\alpha)}{1-\alpha\gamma\lambda_s}, (1-\alpha) + \alpha\gamma\lambda_{s+1}, \ldots, (1-\alpha) + \alpha\gamma\lambda_n$.

Now suppose $E_s = \sum_{i=1}^s \lambda_i u_i u_i^\top$ satisfies Fact 3. Similarly, let $U_s = [u_1, \ldots, u_s]$. Then, $E_s = U_s D_{s,\lambda_i} U_s^\top$. By Schur decomposition, $\mathcal{P}^\pi = URU^\top$ where $R$ is an upper triangular matrix with $R_{ii} = \lambda_i$ for $i = 1, \ldots, n$, and $U$ is a unitary matrix. By simple calculation, we have $\mathcal{P}^\pi U_s = U_s R_s$ where $R_s$ is the $s \times s$ submatrix of $R$ such that $(R_s)_{ij} = R_{ij}$ for $1 \le i, j \le s$ and $(I - \alpha\gamma E_s)^{-1} = I + U_s D_{s,\frac{\gamma\alpha\lambda_i}{1-\gamma\alpha\lambda_i}} U_s^\top$.

Since

$$(\mathcal{P}^\pi - U_s D_{s,\lambda_i} U_s^\top)(U_s D_{s,\frac{\gamma\alpha\lambda_i}{1-\gamma\alpha\lambda_i}} U_s^\top) = U_s R_s D_{s,\frac{\gamma\alpha\lambda_i}{1-\gamma\alpha\lambda_i}} U_s^\top - U_s D_{s,\lambda_i} D_{s,\frac{\gamma\alpha\lambda_i}{1-\gamma\alpha\lambda_i}} U_s^\top$$
$$= U_s (R_s - D_{s,\lambda_i}) D_{s,\frac{\gamma\alpha\lambda_i}{1-\gamma\alpha\lambda_i}} U_s^\top,$$

we get

$$(\mathcal{P}^\pi - E_s)(I - \alpha\gamma E_s)^{-1} = (\mathcal{P}^\pi - U_s D_{s,\lambda_i} U_s^\top)(I + U_s D_{s,\frac{\gamma\alpha\lambda_i}{1-\gamma\alpha\lambda_i}} U_s^\top)$$
$$= (\mathcal{P}^\pi - U_s D_{s,\lambda_i} U_s^\top) + U_s (R_s - D_{s,\lambda_i}) D_{s,\frac{\gamma\alpha\lambda_i}{1-\gamma\alpha\lambda_i}} U_s^\top.$$

Then,

$$(1-\alpha)(I - \alpha\gamma E_s)^{-1} + \alpha\gamma(\mathcal{P}^\pi - E_s)(I - \alpha\gamma E_s)^{-1}$$
$$= (1-\alpha)I + (1-\alpha)U_s D_{s,\frac{\gamma\alpha\lambda_i}{1-\gamma\alpha\lambda_i}} U_s^\top + \alpha\gamma URU^\top - \alpha\gamma U_s D_{s,\lambda_i} U_s^\top + \alpha\gamma U_s (R_s - D_{s,\lambda_i}) D_{s,\frac{\gamma\alpha\lambda_i}{1-\gamma\alpha\lambda_i}} U_s^\top$$
$$= (1-\alpha)I + \alpha\gamma URU^\top + U_s \left( D_{s,\frac{(1-\alpha)\gamma\alpha\lambda_i}{1-\gamma\alpha\lambda_i}} - D_{s,\alpha\gamma\lambda_i} + \alpha\gamma(R_s - D_{s,\lambda_i}) D_{s,\frac{\gamma\alpha\lambda_i}{1-\gamma\alpha\lambda_i}} \right) U_s^\top$$
$$= (1-\alpha)I + \alpha\gamma URU^\top + U_s \left( D_{\frac{(\lambda_i\gamma-1)\gamma\alpha^2\lambda_i}{1-\gamma\alpha\lambda_i}} + \alpha\gamma(R_s - D_{\lambda_i}) D_{\frac{\gamma\alpha\lambda_i}{1-\gamma\alpha\lambda_i}} \right) U_s^\top$$
$$= (1-\alpha)I + \alpha\gamma URU^\top + U_s R_{s,\frac{(\lambda_i\gamma-1)\gamma\alpha^2\lambda_i}{1-\gamma\alpha\lambda_i}} U_s^\top$$
$$= U \left( (1-\alpha)I + \alpha\gamma R + e_{1:s} R_{s,\frac{(\lambda_i\gamma-1)\gamma\alpha^2\lambda_i}{1-\gamma\alpha\lambda_i}} e_{1:s}^\top \right) U^\top,$$

where $R_{s,\frac{(\lambda_i\gamma-1)\gamma\alpha^2\lambda_i}{1-\gamma\alpha\lambda_i}}$ is $s \times s$ upper triangular matrix with diagonal entries $\frac{(\lambda_1\gamma-1)\gamma\alpha^2\lambda_1}{1-\gamma\alpha\lambda_1}, \ldots, \frac{(\lambda_s\gamma-1)\gamma\alpha^2\lambda_s}{1-\gamma\alpha\lambda_s}$, satisfying $R_{s,\frac{(\lambda_i\gamma-1)\gamma\alpha^2\lambda_i}{1-\gamma\alpha\lambda_i}} = D_{s,\frac{(\lambda_i\gamma-1)\gamma\alpha^2\lambda_i}{1-\gamma\alpha\lambda_i}} + \alpha\gamma(R_s - D_{s,\lambda_i})D_{s,\frac{\gamma\alpha\lambda_i}{1-\gamma\alpha\lambda_i}}$. By simple calculation, we know that $(1-\alpha)I + \alpha\gamma R + e_{1:s} R_{\frac{(\alpha\lambda_i\gamma-\alpha)\gamma\alpha\lambda_i}{1-\gamma\alpha\lambda_i}} e_{1:s}^\top$ is $n \times n$ upper triangular matrix with diagonal entries $\frac{(1-\alpha)}{1-\alpha\gamma\lambda_1}, \ldots, \frac{(1-\alpha)}{1-\alpha\gamma\lambda_s}, (1-\alpha) + \alpha\gamma\lambda_{s+1}, \ldots, (1-\alpha) + \alpha\gamma\lambda_n$.

Therefore, by spectral analysis, we conclude that

$$\|V^k - V^\pi\| = \tilde{O}(|\lambda|^k \|V^0 - V^\pi\|)$$

as $k \to \infty$, where

$$\lambda = \max_{1 \le i \le s, s+1 \le j \le n} \left\{ \left| \frac{1-\alpha}{1-\alpha\gamma\lambda_i} \right|, |1 - \alpha + \alpha\gamma\lambda_j| \right\}.$$

## B.2 Proof of Collorary 3.2

Putting $\alpha = 1$ in Theorem 3.1, we conclude Corollary 3.2.

### B.3 Proof of Theorem 4.1

By the definition of DDVI for Control (equation 6 and equation 7),

$$V^k - V^\star = \left(I + \frac{\gamma}{1-\gamma}E_1\right)\left(W^k - (I - \gamma E_1)V^\star\right),$$

where $E_1 = \mathbf{1}v^\top$. Let $W^\star = (I - \gamma E_1)V^\star$. Note that

$$\begin{aligned}
\max_\pi\{r^\pi + \gamma(\mathcal{P}^\pi - E_1)W^\star\} &= \max_\pi\{r^\pi + \gamma(\mathcal{P}^\pi - E_1)V^\star\} \\
&= \max_\pi\{r^\pi + \gamma\mathcal{P}^\pi V^\star\} - \gamma E_1 V^\star \\
&= W^\star.
\end{aligned}$$

Thus,

$$W^k - W^\star = \gamma\left(\mathcal{P}^{\pi_g} - E_1\right)W^{k-1} + r^{\pi_g} - \gamma\left(\mathcal{P}^{\pi^\star} - E_1\right)W^\star - r^{\pi^\star},$$

where $\pi_g = \arg\max_\pi\{r^\pi + \gamma(\mathcal{P}^\pi - E_1)W^{k-1}\}$ and $\pi^\star$ is an optimal policy. Then, by the definition of greedy policy, we have

$$\gamma\left(\mathcal{P}^{\pi_g} - E_1\right)W^{k-1} + r^{\pi_g} - \gamma\left(\mathcal{P}^{\pi^\star} - E_1\right)W^\star - r^{\pi^\star} \leq \gamma\left(\mathcal{P}^{\pi_g} - E_1\right)(W^{k-1} - W^\star),$$

$$\gamma\left(\mathcal{P}^{\pi_g} - E_1\right)W^{k-1} + r^{\pi_g} - \gamma\left(\mathcal{P}^{\pi^\star} - E_1\right)W^\star - r^{\pi^\star} \geq \gamma\left(\mathcal{P}^{\pi^\star} - E_1\right)(W^{k-1} - W^\star).$$

This implies that

$$\begin{aligned}
W^k - W^\star &\leq \gamma\left(\mathcal{P}^{\pi_g} - E_1\right)(W^{k-1} - W^\star), \\
W^k - W^\star &\geq \gamma\left(\mathcal{P}^{\pi^\star} - E_1\right)(W^{k-1} - W^\star),
\end{aligned}$$

and

$$\gamma\mathcal{P}^{\pi^\star}(W^{k-1} - W^\star) \leq W^k - W^\star + \gamma E_1(W^{k-1} - W^\star) \leq \gamma\mathcal{P}^{\pi_g}(W^{k-1} - W^\star).$$

Then, there exist $0 \leq t_i \leq 1$ such that

$$t_i\left(\gamma\mathcal{P}^{\pi^\star}(W^{k-1} - W^\star)\right)_i + (1 - t_i)\left(\gamma\mathcal{P}^{\pi_g}(W^{k-1} - W^\star)\right)_i = \left(W^k - W^\star + \gamma E_1(W^{k-1} - W^\star)\right)_i,$$

for $1 \leq i \leq n$. Define $\pi_k(a \,|\, i) = t_i\pi^\star(a \,|\, i) + (1 - t_i)\pi_g(a \,|\, i)$ for all $a \in \mathcal{A}$ and $1 \leq i \leq n$. Then $\pi_k$ satisfies

$$W^k - W^\star = \gamma\left(\mathcal{P}^{\pi_k} - E_1\right)(W^{k-1} - W^\star).$$

Thus, we get

$$\begin{aligned}
W^k - W^\star &= \gamma^k\prod_{i=1}^{k}\left(\mathcal{P}^{\pi_i} - E_1\right)(W^0 - W^\star) \\
&= \gamma^k\prod_{i=1}^{k}\left(\mathcal{P}^{\pi_i} - E_1\right)(I - \gamma E_1)(V^0 - V^\star) \\
&= \gamma^k\prod_{i=1}^{k}\left(\mathcal{P}^{\pi_i} - E_1\right)(V^0 - V^\star) \\
&= \gamma^k\left(\mathcal{P}^{\pi_k} - E_1\right)\prod_{i=1}^{k-1}\mathcal{P}^{\pi_i}(V^0 - V^\star),
\end{aligned}$$

where the third and last equality follows from $(\mathcal{P}^{\pi_i} - E_1)E_1 = 0$. This implies that

$$V^k - V^\star = \gamma^k \left(I + \frac{\gamma}{1-\gamma}E_1\right)(\mathcal{P}^{\pi_k} - E_1)\prod_{i=1}^{k-1}\mathcal{P}^{\pi_i}(V^0 - V^\star)$$

$$= \gamma^k \left(\mathcal{P}^{\pi_k} + \frac{\gamma}{1-\gamma}E_1\mathcal{P}^{\pi_k} - \frac{1}{1-\gamma}\mathbf{1}v^\top\right)\prod_{i=1}^{k-1}\mathcal{P}^{\pi_i}(V^0 - V^\star).$$

Then, we have

$$\|V^k - V^\star\| \le \frac{2}{1-\gamma}\gamma^k\|V^0 - V^\star\|,$$

since $E_1\mathcal{P}^{\pi_k}$ is stochastic matrix with $\|E_1\mathcal{P}^{\pi_k}\| = 1$.

Suppose there exists a unique optimal policy $\pi^\star$. Then, if (non-deflated) VI generates $V^k$ for $k = 0, 1, \ldots$, there exist $K$ such that $\arg\max_\pi T^\pi V^k = \pi^\star$ for $k > K$. Since DDVI for Control generates the same policy as VI for Control, DDVI for Control also generates greedy policy $\pi^\star$ for $k > K$ iterations. Therefore, we get

$$\|V^k - V^\star\| = \left\|\gamma^k\left(I + \frac{\gamma}{1-\gamma}\mathbf{1}v^\top\right)\left(\mathcal{P}^{\pi^\star} - E_1\right)^{k-K}\prod_{i=1}^{K}(\mathcal{P}^{\pi_i} - E_1)(V^0 - V^\star)\right\|$$

$$= C\gamma^k\left\|\left(\mathcal{P}^{\pi^\star} - E_1\right)^{k-K}\right\|\|V^0 - V^\star\|$$

$$= \tilde{O}(|\gamma\lambda_2|^k\|V^0 - V^\star\|)$$

for $k > K$ and some constant $C \in \mathbb{R}$.

## B.4  Proof of Theorem 5.1

Consider the following stochastic approximation algorithm

$$Y^{k+1} = Y^k + \eta_k(X_k)(X_k)f(Y^k, \zeta_k) \tag{11}$$

for $k = 0, 1, \ldots$, where $Y^k \in \mathbb{R}^n$, $\eta_k(X_k) \in \mathbb{R}^+$, $f(\cdot, z) : \mathbb{R}^n \to \mathbb{R}^n$ is uniformly Lipschitz, and $\{\zeta_k\}_{k=0,1,\ldots}$ are i.i.d. random variables. Let $F(y) = \mathbf{E}[f(y, \zeta_k)]$, $F_\infty(y) = \lim_{r\to\infty} F(ry)/r$, $M^{k+1} = f(Y^k, \zeta_k) - F(Y^t)$, and $\mathcal{F}^k = \sigma(Y^i, M^i, 1 \le i \le t)$. The following result from Borkar & Meyn (2000) shows the convergence of the $Y^k$ sequence.

**Proposition B.1.** *(Borkar & Meyn, 2000, Theorem 2.2 and 2.5) If (i) $\eta_k = (k+1)^{-1}$, (ii) $\{M^k, \mathcal{F}^k\}_{k=0,1,\ldots}$, are martingale difference sequence , (iii) $\mathbf{E}[M^{k+1}|\mathcal{F}^k] \le K(1 + \|Y^k\|^2)$ for some constant $K$, (iv) $\dot{y}(t) = F_\infty(y(t))$ has asymptotically stable equilibrium origin, and (v) $\dot{y}(t) = F(y(t))$ has a unique globally asymptotically stable equilibrium $y^\star$, then $\{Y^t\}_{k=0,1,\ldots}$ of equation 11 converges to $y^\star$ almost surely, and furthermore, $\{Y^t\}_{k=0,1,\ldots}$ of*

$$Y^{k+1}(i_k) = Y^k(i_k) + \eta_{\nu_{i_k,k}}f(Y^k, \zeta_k)(i_k) \tag{12}$$

*for $k = 0, 1, \ldots$, where $i_k$ are random variables taking values in $\{1, 2, \cdots, n\}$ and $\nu_{i_k,k} = \sum_{t=0}^{k}\mathbf{1}_{\{i=i_t\}}$ when $i_k = i$ satisfying $\liminf \nu_{i_k,k}/k > c$ for some constant $c > 0$, also converge to $y^\star$ almost surely.*

In equation 12, $Y^k(i_k)$ and $f(Y^k, \zeta_k)(i_k)$ are the $i$-th coordinate of $Y^k$ and $f(Y^k, \zeta_k)$, respectively. The equation 12 can be interpreted as asynchronous version of equation 11. We note that Proposition B.1 is a simplified version with stronger conditions of Theorem 2.2 and 2.5 of Borkar & Meyn (2000).

Recall that DDTD equation 8 with $\alpha = 1$ is

$$W^{k+1}(X_k) = W^k(X_k) + \eta_k(X_k)\left[r^\pi(X_k) + \gamma V^k(X'_k) - \gamma(E_sV^k)(X_k) - W^k(X_k)\right]$$

$$V^{k+1} = (I - \gamma E_s)^{-1}W^{k+1}. \tag{13}$$

We will first show that $W^k \to (I - \gamma E_s)V^\pi$ and this directly implies that $V^k \to V^\pi$. For applying Proposition B.1, consider

$$W^{k+1} = W^k + \eta_k f(W^k, Z_t),$$

where $\eta_k = (k+1)^{-1}$, $Z_t \in \mathbb{R}^n$ such that $Z_t(x) \sim \mathcal{P}^\pi(\cdot \,|\, x)$ and $\{Z_t\}_{k=0,1,\dots}$ are i.i.d. random variables, and $f(W^k, Z_t)(x) = r^\pi(x) + \gamma((I - \gamma E_s)^{-1} W^k)(Z_t(x)) - \gamma E_s (I - \gamma E_s)^{-1} W^k(x) - W^k(x)$. Then,

$$F(W) = \mathbf{E}[f(W, Z_t)] = [\gamma(\mathcal{P}^\pi - E_s)(I - \gamma E_s)^{-1} - I]W + r^\pi,$$

$M^{k+1} = f(W^k, Z_t) - F(W^k)$, and $\mathcal{F}^k = \sigma(W^i, M^i, 1 \le i \le t)$.

We now check the conditions. First, since $f(W, z)(x) - f(W', z)(x) = (I - \gamma E_s)^{-1}(W - W')(z(x)) - (I + \gamma E_s (I - \gamma E_s)^{-1})(W - W')(x)$ implies $\|f(W, z) - f(W', z)\| \le (\| (I - \gamma E_s)^{-1} \| + \|(I + \gamma E_s (I - \gamma E_s)^{-1})\|)\|W - W'\|$, $f(\cdot, z)$ is uniformly Lipschitz. We have

$$\mathbf{E}[M^{k+1}(x) \,|\, \mathcal{F}_t]$$
$$= \mathbf{E}[\gamma((I - \gamma E_s)^{-1} W^k)(Z_t(x)) - \gamma(E_s (I - \gamma E_s)^{-1} W^k)(x) - \gamma((\mathcal{P}^\pi - E_s))(I - \gamma E_s)^{-1}W^k)(x) \,|\, \mathcal{F}^k]$$
$$= \gamma\mathcal{P}^\pi (I - \gamma E_s)^{-1} W^k)(x) - \gamma\mathcal{P}^\pi(I - \gamma E_s)^{-1}W^k(x)$$
$$= 0$$

for all $x \in \mathcal{X}$. This implies that $M^{k+1}$ is a martingale difference sequence. Also,

$$\mathbf{E}[\|M^{k+1}\|^2 \,|\, \mathcal{F}^k] \le (\|\gamma (I - \gamma E_s)^{-1} \| + \|\gamma\mathcal{P}^\pi(I - \gamma E_s)^{-1}\|)^2 \|W^k\|^2 \le K(1 + \|W^k\|^2)$$

for constant $K = (\|\gamma (I - \gamma E_s)^{-1} \| + \|\gamma\mathcal{P}^\pi(I - \gamma E_s)^{-1}\|)^2$.

By definition, $F_\infty(y) = \lim_{r \to \infty} F(ry)/r = \gamma(\mathcal{P}^\pi - E_s))(I - \gamma E_s)^{-1} - I]y$. In Section B.1, we showed that $\mathrm{spec}(F_\infty + I) = \{\gamma\lambda_{s+1}, \dots, \gamma\lambda_n, 0\}$. Hence, $\mathrm{spec}(F_\infty) = \{\gamma\lambda_{s+1} - 1, \dots, \gamma\lambda_1 - 1, -1\}$. Since $|\lambda_i| \le 1$ for all $s + 1 \le i \le n$, real parts of all eigenvalues $F_\infty$ are negative. This implies that $\dot{y}(t) = F_\infty(y(t))$ has an asymptotically stable equilibrium origin and $\dot{y}(t) = F(y(t))$ has a unique globally asymptotically stable equilibrium $W^\pi = [I - (\gamma(\mathcal{P}^\pi - E_s))(I - \gamma E_s)^{-1}]^{-1}r^\pi$.

Lastly, since $X_k \sim \mathrm{Unif}(\mathcal{X})$ and $\{X_k\}_{k=0,1,\dots}$ are i.i.d. random variables, $\nu_{i,t}/t$ converges to $\frac{1}{n}$ almost surely by the law of large numbers. Since $\eta_{\nu_{X_k}, k} = (\sum_{i=0}^k \mathbf{1}_{X_k = X_i})^{-1}$, Therefore, by Proposition B.1, $\{W^k\}_{k=0,1,\dots}$ of equation 13 converge to $W^\pi$ almost surely, and this implies that $V^k \to (I - \gamma E_s)^{-1}W^\pi = (I - \gamma\mathcal{P}^\pi)^{-1}r^\pi = V^\pi$ almost surely.

### B.5 Proof of Theorem 5.2

Consider the following linear stochastic approximation algorithm

$$Y^{k+1} = Y^k + \eta_k(X_k)(A(\zeta_k)Y^t + b(\zeta_k)) \tag{14}$$

for $k = 0, 1, \dots$, where $Y^k \in \mathbb{R}^n$, $\{\zeta_k\}_{k=0,1,\dots}$, are random variables, $b(\zeta_k) \in \mathbb{R}^n$, $A(\zeta_k) \in \mathbb{R}^n \times \mathbb{R}^n$, and $\|A_{i,j}\|, \|b_k\| < \infty$, for $1 \le i, j \le n$ and $1 \le k \le n$. The following result from Chen et al. (2020) provides a rate of convergence of $Y^k$ to the solution $Y^\star = -A^{-1}b$, where $A$ and $b$ are the expectation of matrices $A(\zeta_k)$ and vectors $b(\zeta_k)$, respectively.

**Proposition B.2.** *(Chen et al., 2020, Theorem 2.5 and Corollary 2.7) If (i) $\eta_k = g(k + 1)^{-1}$ for $g > 0$, (ii) $\{\zeta_k\}_{k=0,1,\dots}$ is ergodic (aperiodic and irreducible) Markov process with a unique invariant measure $\mu$, (iii) $\mathbf{E}_\mu[A^t] = A$ and $\mathbf{E}_\mu[B^t] = b$, (iv) $A$ is Hurwitz, (v) $Re(\lambda) < -1/2$ for all $\lambda \in spec(gA)$, then $\mathbf{E}[\|Y^t - Y^\star\|^2] = O(k^{-1})$ where $Y^\star = -A^{-1}b$.*

We note that Proposition B.2 is also a simplified version with stronger conditions of Theorem 2.5 and Corollary 2.7 of Chen et al. (2020).

Define $\psi(X_k) \in \mathbb{R}^n$ as $\psi(X_k)(x) = \mathbf{1}_{\{X_k = x\}}$. Then, DDTD equation 8 with $\alpha = 1$ is equivalent to

$$W^{k+1} = W^k + \eta_k(X_k)(A(X_k, X_k')W^k + b(X_k, X_k')),$$

where

$$A(X, X') = \psi(X)(\psi(X')^\top \gamma (I - \gamma E_s)^{-1} - \psi(X)^\top \gamma E_s (I - \gamma E_s)^{-1} - \psi(X)^\top)$$

and $b(X, X') = \psi(X)r^\pi(X)$. Hence $\|A_{i,j}\|, \|b_k\| < \infty$, for $1 \le i, j \le n$ and $1 \le k \le n$.

Since $\{X_k, X_k'\}_{k=0,1,\dots}$ are i.i.d. random variables, $\{X_k, X_k'\}_{k=0,1,\dots}$ are Markov process. Let $\mu \sim \{X_k, X_k'\}$ and $\mathcal{P}(x_1, x_1' \,|\, x_2, x_2')$ be transition matrix of $\{X_k, X_k'\}_{k=0,1,\dots}$. Then $\mathcal{P}(x_1, x_1' \,|\, x_2, x_2') = \mu(x_1, x_1') = \frac{1}{n}\mathcal{P}^\pi(x_1' \,|\, x_1)$. Since $(\mu')^\top \mathcal{P}(\cdot, \cdot \,|\, \cdot, \cdot) = \mu$ for any distribution $\mu'$, $\mu$ is a unique invariant measure. Hence, $\{X_k, X_k'\}_{k=0,1,\dots}$ is irreducible and aperiodic. We have,

$$\begin{aligned}
\mathbf{E}_\mu[A(X_k, X_k')] &= \mathbf{E}_\mu[\psi(X_k)(\psi(X_k')^\top \gamma (I - \gamma E_s)^{-1} - \psi(X_t)^\top \gamma E_s (I - \gamma E_s)^{-1} - \psi(X_k)^\top) \,|\, X_k] \\
&= \mathbf{E}_\mu[\psi(X_k)(\psi(X_k)^\top \mathcal{P}^\pi (I - \gamma E_s)^{-1} - \psi(X_t)^\top \gamma E_s (I - \gamma E_s)^{-1} - \psi(X_k)^\top)] \\
&= \mathbf{E}_\mu[\psi(X_k)\psi(X_k)^\top (\mathcal{P}^\pi (I - \gamma E_s)^{-1} - \gamma E_s (I - \gamma E_s)^{-1} - I)] \\
&= n^{-1}((\mathcal{P}^\pi - \gamma E_s)(I - \gamma E_s)^{-1} - I),
\end{aligned}$$

and $\mathbf{E}_\mu[b(X_k, X_k')] = n^{-1}r^\pi$. Then, $Y^\star = -A^{-1}b = [I - (\gamma(\mathcal{P}^\pi - E_s))(I - \gamma E_s)^{-1}]^{-1}r^\pi = W^\pi$.

As we discussed in Section B.4, $\mathrm{spec}(A) = \{\gamma\lambda_{s+1} - 1, \dots, \gamma\lambda_n - 1, -1\}$. Let $\lambda_{\mathrm{DDTD}} = \min_{\lambda \in \{\lambda_{s+1},\dots,\lambda_n\}} \mathrm{Re}(1 - \gamma\lambda)$. Then, if $g > \frac{n}{2\lambda_{\mathrm{DDTD}}}$, the real component of the eigenvalues $Re(\lambda) < -1/2$ for all $\lambda \in \mathrm{spec}(gA)$. Therefore, by Proposition B.2, $\mathbf{E}[\|W^k - W^\pi\|^2] = O(k^{-1})$ and this implies that $\mathbf{E}[\|V^k - V^\pi\|^2] \le \mathbf{E}[\|(I - \gamma E_s)^{-1}\|^2\|W^k - W^\star\|^2] = O(k^{-1})$.

### B.6 Non-asymptotic convergence analysis of DDTD

Consider following the stochastic approximation algorithm

$$Y^{k+1} = Y^k + \eta_k(X_k)(f(Y^k, \zeta_k) - Y^k + Z_k) \tag{15}$$

for $k = 0, 1, \dots$, where $Y^k \in \mathbb{R}^n$, $\eta_k(X_k) \in \mathbb{R}^+$, $\{\zeta_k\}_{k=0,1,\dots}$ are Markov chain with unique distribution $\mu$ and transition matrix $P$. Let $F(y) = \mathbf{E}[f(y, \zeta_k)]$ and $\mathcal{F}^k = \sigma(Y^i, M^i, Z_i, \eta_i 1 \le i \le t)$. Define mixing time as $t_\delta = \{\min k \ge 0, : \max_y \|P^k(\cdot, y) - \mu(\cdot)\|_{\mathrm{TV}} < \delta\}$ where $\|\cdot\|_{\mathrm{TV}}$ stands for the total variation distance. Let $Y^\star$ be fixed point of $F(y)$. Then following proposition holds.

**Proposition B.3.** *(Chen et al., 2023, Theorem 1) Let (i) $\{\zeta_k\}_{k=0,1,\dots}$ is aperiodic and irreducible Markov chain, (ii) $F$ is $\beta$-contraction respect to some $\|\cdot\|_c$, (iii) $f(\cdot, z): \mathbb{R}^n \to \mathbb{R}^n$ is $A_1$-uniformly Lipschitz with respect to $\|\cdot\|_c$ and $\|f(0, y)\|_c \le B_1$ for any $y$, (iv) $\{M^k, \mathcal{F}^k\}_{k=0,1,\dots,}$ are martingale difference sequence, and (v) $\mathbf{E}[Z^{k+1} \,|\, \mathcal{F}^k] \le A_2 + B_2\|Y^k\|^2$ for some $A_2, B_2 > 0$. Then, if $\eta_k = \eta$,*

$$\mathbf{E}[\|Y^k - Y^\star\|_c^2] \le c_1 d_1(1 - d_2\eta)^{k - t_\alpha} + \frac{d_3}{d_2}\eta t_\alpha$$

*for all $k \ge K$, and if $\eta_k = \frac{1}{d_2(k+h)}$, we have*

$$\mathbf{E}[\|Y^k - Y^\star\|_c^2] \le c_1 d_1 \frac{K + h}{k + h} + \frac{8\eta^2 d_3 c_2 t_k \log(k + h)}{k + h}$$

*for all $k \ge K$, where $\{\eta_k(X_k)\}_{k=0,1,\dots}$ is nonincreasing sequence satisfying $\sum_{i=k-t_k}^{k-1} \alpha_i \le \min\left\{\frac{d_2}{d_3 A^2}, \frac{1}{4A}\right\}$ for all $k \ge t_k$, $K = \min\{k \ge 0 : k \ge t_k\}$, $A = A_1 + A_2 + 1$, $B = B_1 + B_2$, $c_1 = (\|Y^0 - Y^\star\|_c + \|Y^0\|_c + A/B)^2$, $c_2 = (A\|Y^\star\|_c + B)^2$, $l_{cs}\|\cdot\|_s \le \|\cdot\|_c \le u_{cs}\|\cdot\|_s$ for chosen norm $\|\cdot\|_s$ such that $\frac{1}{2}\|\cdot\|_s^2$ is $L$-smooth function with respect to $\|\cdot\|_s$, $d_1 = \frac{1 + \theta u_{cs}^2}{1 + \theta l_{cs}^2}$, $d_2 = 1 - \beta d_1^{1/2}$, and $d_3 = \frac{114 L(1 + \theta u_{cs}^2)}{\theta l_{cs}^2}$ such that $\theta$ is chosen satisfying $\beta^2 \le (d_1)^{-1}$.*

DDTD in equation 8 with $\alpha = 1$ is equivalent to

$$W^{k+1} = W^k + \eta_k(X_k)(f(W^k, X_k, X_k') - W^k + Z_k),$$

where $Z_k = 0$, and $f(W^k, X, X')(x) = \mathbf{1}_{\{X=x\}}(r^\pi(x) + \gamma((I - \gamma E_s)^{-1} W^k)(X') - \gamma E_s (I - \gamma E_s)^{-1} W^k(x) - W^k(x)) + W^k(x)$. As we showed in Section B.5, $\{X_k, X_k'\}_{k=0,1,\ldots}$ is irreducible and aperiodic Markov chain with an invariant measure $\mu$. Furthermore, mixing time of $\{X_k, X_k'\}_{k=0,1,\ldots}$ is 1.

We also have

$$
\begin{aligned}
F(W) &= \mathbf{E}_{\mu \sim \{X, X'\}}[\mathbf{1}_{\{X=x\}}(r^\pi(x) + \gamma((I - \gamma E_s)^{-1} W)(X') - \gamma E_s (I - \gamma E_s)^{-1} W(x) - W(x)) + W(x)] \\
&= \mathbf{E}_{\mu \sim \{X, X'\}}[\mathbf{1}_{\{X=x\}}(r^\pi(x) + \gamma((I - \gamma E_s)^{-1} W)(X') - \gamma E_s (I - \gamma E_s)^{-1} W(x) - W(x)) + W(x) \mid X] \\
&= \mathbf{E}_{\mu \sim \{X, X'\}}[\mathbf{1}_{\{X=x\}}(r^\pi(x) + \mathcal{P}^\pi \gamma((I - \gamma E_s)^{-1} W)(X) - \gamma E_s (I - \gamma E_s)^{-1} W(x) - W(x)) + W(x)] \\
&= \frac{1}{n}(r^\pi(x) + \gamma(\mathcal{P}^\pi - E_s)(I - \gamma E_s)^{-1} W(x)) + (1 - \frac{1}{n})W(x). \quad (16)
\end{aligned}
$$

Let $\lambda_1, \ldots, \lambda_n$ be the eigenvalues of $\mathcal{P}^\pi$ sorted in the decreasing order of magnitude with ties broken arbitrarily. Let $A_{\text{DDTD}} = \gamma(\mathcal{P}^\pi - E_s)(I - \gamma E_s)^{-1}$. Since spectral radius is the infimum of matrix norm, for any $\epsilon > 0$, there exists $\|\cdot\|_\epsilon$ such that $\rho(\frac{1}{n}A_{\text{DDTD}} + (1 - \frac{1}{n})I) \leq \|\frac{1}{n}A_{\text{DDTD}} + (1 - \frac{1}{n})I\|_\epsilon \leq \rho(\frac{1}{n}A_{\text{DDTD}} + (1 - \frac{1}{n})I) + \epsilon$ where $\rho(\frac{1}{n}A_{\text{DDTD}} + (1 - \frac{1}{n})I) = \max_{\lambda \in \{\lambda_{s+1}, \ldots, \lambda_n\}}\{|1 - \frac{1}{n} + \frac{\gamma}{n}\lambda|\}$.

The upper bound $\|f(0, X, X')\|_\epsilon = \|\mathbf{1}_{x=X} r^\pi(x)\|_\epsilon \leq \|r^\pi\|_\epsilon = B_\epsilon$ and $f(W, X, X')(x) - f(W', X, X')(x) = \mathbf{1}_{\{X=x\}}(\gamma(I - \gamma E_s)^{-1}(W - W')(X') - (\gamma E_s (I - \gamma E_s)^{-1}(W - W'))(x))$ implies that

$$
\begin{aligned}
\|f(W, X, X') - f(W', X, X')\|_{\epsilon'} &\leq \left(\|(\gamma(I - \gamma E_s)^{-1}\|_\epsilon + \|(\gamma E_s (I - \gamma E_s)^{-1}\|_\epsilon\right)\|W - W'\|_\epsilon \\
&\leq A_\epsilon\|W - W'\|_\epsilon.
\end{aligned}
$$

As shown in equation 16, we have $F(W) = \mathbf{E}_{\mu \sim \{X, X'\}}[f(W, X, X')] = \frac{1}{n}(r^\pi + \gamma(\mathcal{P}^\pi - E_s)(I - \gamma E_s)^{-1} W) + (1 - \frac{1}{n})W$. This shows that

$$\|F(W_1) - F(W_2)\|_\epsilon = \left\|\left(\frac{1}{n}A_{\text{DDTD}} + (1 - \frac{1}{n})I\right)(W_1 - W_2)\right\|_\epsilon \leq \left(\frac{1}{n}\rho + \epsilon + (1 - \frac{1}{n})\right)\|W_1 - W_2\|_\epsilon$$

and $F(W)$ has fixed point $W^\pi$ since $\frac{1}{n}A_{\text{DDTD}} + (1 - \frac{1}{n})I$ and $A_{\text{DDTD}}$ share same fixed point.

Thus, by Theorem B.3 and $\mathbf{E}[\|V^k - V^\pi\|_\epsilon^2] \leq \|(I - \gamma E_s)^{-1}\|_\epsilon^2 \mathbf{E}[\|W^k - W^\star\|_\epsilon^2]$, we have the following two non-asymptotic convergence results for DDTD – one for constant stepsize and the other for a diminishing stepsize.

**Theorem B.4** (DDTD with constant stepsize). *Let $\lambda_1, \ldots, \lambda_n$ be the eigenvalues of $\mathcal{P}^\pi$ sorted in decreasing order of magnitude with ties broken arbitrarily. Let $\eta_k = \eta$. For any $\epsilon > 0$, there exist $a_\epsilon, b_\epsilon, c_\epsilon, d_\epsilon > 0$ and $\|\cdot\|_\epsilon$ such that for $\alpha = 1$ and $0 < \eta < d_\epsilon$, DDTD exhibits the rate*

$$\mathbf{E}[\|V^k - V^\pi\|_\epsilon^2] \leq a_\epsilon(1 - \eta + \rho b_\epsilon \eta)^{k-1} + c_\epsilon \eta$$

*for all $k \geq 1$ where $\rho = 1 - \frac{1}{n} + \frac{\gamma}{n}\max\{|\lambda_{s+1}|, \ldots, |\lambda_n|\} + \epsilon$.*

We note that $a_\epsilon, b_\epsilon, c_\epsilon, d_\epsilon$ of Theorem B.4 can be obtained by plugging $A_1 = A_\epsilon, A_2 = 0, B_1 = B_\epsilon, B_2 = 0$ in Proposition B.3.

**Theorem B.5** (DDTD with diminishing stepsize). *Let $\lambda_1, \ldots, \lambda_n$ be the eigenvalues of $\mathcal{P}^\pi$ sorted in decreasing order of magnitude with ties broken arbitrarily. For any $\epsilon > 0$, there exist $a_\epsilon, b_\epsilon, c_\epsilon, d_\epsilon > 0$ and $\|\cdot\|_\epsilon$ such that for $\alpha = 1$ and $\eta_k(X_k) = \frac{1}{(k+b_\epsilon)(1-c_\epsilon\rho)}$, DDTD exhibits the rate*

$$\mathbf{E}[\|V^k - V^\pi\|_\epsilon^2] \leq \frac{a_\epsilon}{k + b_\epsilon} + \left(\frac{1}{1 - c_\epsilon \rho}\right)^2 \frac{d_\epsilon}{k + b_\epsilon}$$

*for all $k \geq 1$ where $\rho = 1 - \frac{1}{n} + \frac{\gamma}{n}\max\{|\lambda_{s+1}|, \ldots, |\lambda_n|\} + \epsilon$.*

Again, we note that $a, b, c,$ of Theorem B.5 can be obtained by plugging $A_1 = A_\epsilon, A_2 = 0, B_1 = B_\epsilon, B_2 = 0$ in Proposition B.3.

---

**Algorithm 4** DDVI with AutoPI (Detailed)

---

1: Initialize $C, \epsilon$
2: **function** DDVI($(s, V, K, \{\lambda_i\}_{i=1}^s, \{u_i\}_{i=1}^s, \alpha_s)$)
3:     $V^0 = V$, $c = 0$
4:     $E_s = \sum_{i=1}^s \lambda_i u_i v_i^\top$ as in Fact 2
5:     **for** $k = 0, \ldots, K-1$ **do**
6:         **if** c $\geq C$ and $\left| \frac{w^{k+1}}{\|w^{k+1}\|_2} - \frac{w^k}{\|w^k\|_2} \right| < \epsilon$ **then**
7:             $\lambda'_{s+1} = (w^k)^\top w^{k+1}/\|w^k\|_2^2$
8:             $\lambda_{s+1} = (\lambda'_{s+1} - 1 + \alpha_s)/(\alpha_s \gamma)$
9:             $u_{s+1} = w^{k+1} - \sum_{i=1}^s \frac{\alpha \lambda_i (1-\gamma \lambda_i)}{1-\alpha_s \gamma \lambda_i} \frac{v_i^\top w^{k+1}}{\lambda_i - \lambda_{s+1}} u_i$
10:             Return
                  DDVI($s+1, V^k, K-c, \{\lambda_i\}_{i=1}^{s+1}, \{u_i\}_{i=1}^{s+1}, \alpha_{s+1}$)
11:         **else**
12:             $W^{k+1} = (1-\alpha_s)V^k + \alpha_s \gamma (\mathcal{P}^\pi - \alpha_s E_s)V^k + \alpha_s r^\pi$
13:             $V^{k+1} = (I - \alpha_s \gamma E_s)^{-1} W^{k+1}$
14:             c = c+1
15:             $w^{k+1} = W^{k+1} - W^k$
16:             $w^k = W^k - W^{k-1}$
17:     Return $V^K$
18: Initialize $V^0$ and $u_1 = \mathbf{1}$, $\lambda_1 = 1$
19: DDVI($1, V^0, K, \lambda_1, u_1, 1$)

---

# C    DDVI with QR Iteration, AutoPI and AutoQR

## C.1   DDVI with AutoPI and AutoQR

We first introduce DDVI with AutoPI for $0 < \alpha \leq 1$. If $W^0 = 0$, we have

$$W^k = \sum_{i=0}^{k-1} ((1-\alpha)(I - \alpha \gamma E_s)^{-1} + \alpha \gamma (\mathcal{P}^\pi - E_s)(I - \alpha \gamma E_s)^{-1})^i \alpha r^\pi$$

by equation 4, and

$$(W^{k+1} - W^k) = ((1-\alpha)(I - \alpha \gamma E_s)^{-1} + \alpha \gamma (\mathcal{P}^\pi - E_s)(I - \alpha \gamma E_s)^{-1})^k \alpha r^\pi$$

are the iterates of a power iteration with respect to $(1-\alpha)(I - \alpha \gamma E_s)^{-1} + \alpha \gamma (\mathcal{P}^\pi - E_s)(I - \alpha \gamma E_s)^{-1}$. In Section B.1, we show that

$$(1-\alpha)(I - \alpha \gamma E_s)^{-1} + \alpha \gamma (\mathcal{P}^\pi - E_s)(I - \alpha \gamma E_s)^{-1} = (1-\alpha)I + \alpha \gamma \mathcal{P}^\pi + U_s D_{s, \frac{(\lambda_i \gamma - 1)\gamma \alpha^2 \lambda_i}{1-\gamma \alpha \lambda_i}} V_s^\top.$$

For $\alpha \approx 1$, we expect that spectral radius of matrix be $1 - \alpha + \gamma \alpha \lambda_{s+1}$ and $(1-\alpha)I + \alpha \gamma \mathcal{P}^\pi + U_s D_{s, \frac{(\lambda_i \gamma - 1)\gamma \alpha^2 \lambda_i}{1-\gamma \alpha \lambda_i}} V_s^\top$ and $\mathcal{P}^\pi + U_s D_{s, \frac{(\lambda_i \gamma - 1)\alpha \lambda_i}{1-\gamma \alpha \lambda_i}} V_s^\top$ have the same top eigenvector. With the same argument as in Section 4.2, we can recover the $s+1$-th eigenvector of $\mathcal{P}^\pi$ by Bru et al. (2012, Proposition 5). Leveraging this observation, we formalize this approach in Algorithm 4.

Now, we introduce DDVI with AutoQR for $0 < \alpha \leq 1$.

Assume the top $s+1$ eigenvalues of $\mathcal{P}^\pi$ are distinct, i.e., $1 = \lambda_1 > |\lambda_2| > \cdots > |\lambda_{s+1}|$. Let $E_s = \sum_{i=1}^s \lambda_i u_i u_i^\top$ be a rank-$s$ deflation matrix of $\mathcal{P}^\pi$ as in Fact 3. If $W^0 = 0$, we again have

$$W^k = \sum_{i=0}^{k-1} ((1-\alpha)(I - \alpha \gamma E_s)^{-1} + \alpha \gamma (\mathcal{P}^\pi - E_s)(I - \alpha \gamma E_s)^{-1})^i \alpha r^\pi,$$

$$W^{k+1} - W^k = ((1-\alpha)(I - \alpha \gamma E_s)^{-1} + \alpha \gamma (\mathcal{P}^\pi - E_s)(I - \alpha \gamma E_s)^{-1})^k \alpha r^\pi,$$

---

**Algorithm 5** DDVI with AutoQR

---

1: Initialize $C, \epsilon$
2: **function** DDVI$((s, V, K, \{\lambda_i\}_{i=1}^s, \{u_i\}_{i=1}^s, \alpha_s))$
3:     $V^0 = V$, $c = 0$,
4:     $E_s = \sum_{i=1}^s \lambda_i u_i u_i^\top$ as in Fact 3
5:     **for** $k = 0, \ldots, K-1$ **do**
6:         **if** c $\geq C$ and $\left| \frac{w^{k+1}}{\|w^{k+1}\|_2} - \frac{w^k}{\|w^k\|_2} \right| < \epsilon$ **then**
7:             $\lambda'_{s+1} = (w^k)^\top w^{k+1} / \|w^k\|_2^2$
8:             $\lambda_{s+1} = (\lambda'_{s+1} - 1 + \alpha_s)/(\alpha_s \gamma)$
9:             $u'_{s+1} = w^{k+1} - \sum_{i=1}^s u_i^\top w^{k+1} u_i$
10:            $u_{s+1} = \frac{1}{\sqrt{(u'_{s+1})^\top u'_{s+1}}} u'_{s+1}$
11:            Return
                DDVI$(s+1, V^k, K-c, \{\lambda_i\}_{i=1}^{s+1}, \{u_i\}_{i=1}^{s+1}, \alpha_s)$
12:         **else**
13:             $W^{k+1} = (1-\alpha_s)V^k + \alpha_s\gamma(\mathcal{P}^\pi - \alpha_s E_s)V^k + \alpha_s r^\pi$
14:             $V^{k+1} = (I - \alpha_s\gamma E_s)^{-1} W^{k+1}$
15:             c = c+1
16:             $w^{k+1} = W^{k+1} - W^k$
17:             $w^k = W^k - W^{k-1}$
18:     Return $V^K$
19: Initialize $V^0$ and $u_1 = \mathbf{1}$, $\lambda_1 = 1$
20: DDVI$(1, V^0, K, \lambda_1, u_1, 1)$

---

and

$$(W^{k+1} - W^k) = ((1-\alpha)(I - \alpha\gamma E_s)^{-1} + \alpha\gamma(\mathcal{P}^\pi - E_s)(I - \alpha\gamma E_s)^{-1})^k \alpha r^\pi$$

is the iterates of a power iteration with respect to $(1-\alpha)(I - \alpha\gamma E_s)^{-1} + \alpha\gamma(\mathcal{P}^\pi - E_s)(I - \alpha\gamma E_s)^{-1}$. In Section B.1, we show that

$$(1-\alpha)(I - \alpha\gamma E_s)^{-1} + \alpha\gamma(\mathcal{P}^\pi - E_s)(I - \alpha\gamma E_s)^{-1} = (1-\alpha)I + \alpha\gamma\mathcal{P}^\pi + U_s R_{s, \frac{(\lambda_i\gamma-1)\gamma\alpha^2\lambda_i}{1-\gamma\alpha\lambda_i}} U_s^\top.$$

For $\alpha \approx 1$, we expect that spectral radius of matrix is $1-\alpha+\gamma\alpha\lambda_{s+1}$ and $(1-\alpha)I + \alpha\gamma\mathcal{P}^\pi + U_s R_{s, \frac{(\lambda_i\gamma-1)\gamma\alpha^2\lambda_i}{1-\gamma\alpha\lambda_i}} V_s^\top$ and $\mathcal{P}^\pi + U_s R_{s, \frac{(\lambda_i\gamma-1)\alpha\lambda_i}{1-\gamma\alpha\lambda_i}} U_s^\top$ have the same top eigenvector. With the same argument as in Section 4.2, for large $k$, we expect $\frac{W^{k+1}-W^k}{\|W^{k+1}-W^k\|} \approx w$, where $w$ is the top eigenvector of $\mathcal{P}^\pi - U_s R_{s, \frac{(\lambda_i\gamma-1)\alpha\lambda_i}{1-\gamma\alpha\lambda_i}} U_s^\top$. Thus, by orthornormalizing $w$ against $u_1, \ldots, u_s$, we can obtain top $s+1$-th Schur vector of $\mathcal{P}^\pi$ Saad (2011, Section 4.2.4). Leveraging this observation, we propose DDVI with Automatic QR Iteration (AutoQR) and formalize this in Algorithm 5.

# D   Environments

We introduce four environments and polices used in experiments.

**Cliffwalk**   The Cliffwalk is a $3 \times 7$ grid world. The top-left corner is initial state, top-right corner is the goal terminal state with reward of 10, and the other states in the first row are terminal states with a reward of $-10$. There is a penalty of $-1$ in all other states. The agent has four actions: UP(0), RIGHT(1), DOWN(2), and LEFT(3). Each action has 90% chance to successfully move in the selected direction, and with probability of 10% one of the other three directions is randomly selected and the agent moves in that direction. If the agent attempts to get out of the boundary, it will stay in place. We use the optimal policy for the policy evaluation task in our experiments. The environment is shown in Figure 4.

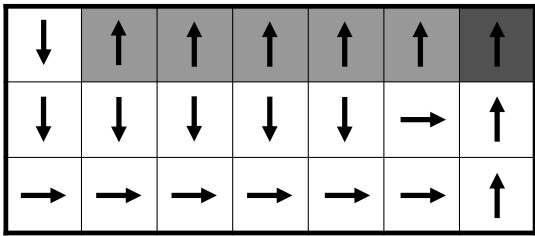

Figure 4: Cliffwalk environment. Dark shaded region is goal state with a positive reward. Gray shaded regions are terminal states with negative reward. Arrows show the optimal policy.

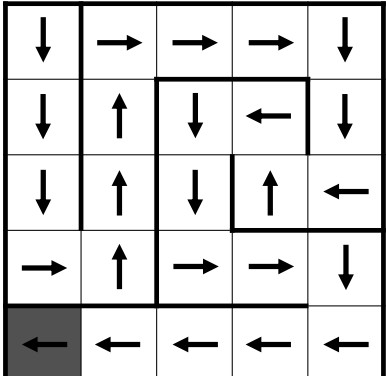

Figure 5: Maze environment. The dark shaded region is the goal state with positive reward. Arrows show optimal policy.

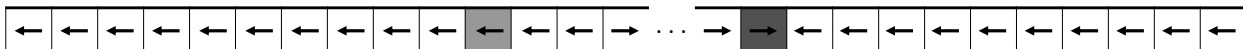

Figure 6: Chain walk environment. Dark shaded region is the goal state with a positive reward and gray shaded region is state with a negative reward. Arrows show optimal policy.

**Maze**  The Maze is a $5 \times 5$ grid world with 16 walls. The top-left corner is the initial state, and the bottom-left corner is the goal state with reward of 10. Similar to the Clifffwalk, there is a penalty of $-1$ in all other states, and the agent has four actions: UP(0), RIGHT(1), DOWN(2), and LEFT(3). Each action has 90% chance to successfully move in the selected direction, and with probability of 10% one of the other three directions is randomly selected and the agent moves in that direction. If the agent attempts to get out of the boundary or hits a wall, it will stay in place. We use the following policies for the policy evaluation task in our DDVI and DDTD experiments, respectively:

$$(2, 2, 3, 0, 3, 0, 2, 1, 3, 2, 2, 2, 3, 3, 1, 0, 3, 0, 3, 3, 2, 2, 1, 1, 0),$$
$$(2, 2, 3, 0, 3, 0, 2, 1, 3, 2, 2, 2, 3, 3, 1, 0, 3, 0, 3, 3, 3, 3, 1, 1, 0).$$

Here, the action for each state is listed row-wise. The environment is shown in Figure 5.

**Chain Walk**  We use the Chain Walk environment, as described by Farahmand & Ghavamzadeh (2021), which is similar to the formulation by Lagoudakis & Parr (2003). Chain Walk is parametrized by the tuple $(|\mathcal{X}|, |\mathcal{A}|, b_p, b_r)$. It is a circular chain, where the state 1 and 50 are connected. The reward is state-dependent. State 40 is goal state with reward 1, state 11 gives reward $-1$, and all other states give reward 0. Agent has two actions: RIGHT(0) and LEFT(1). With each action, the agent has 70% chance to successfully move in the selected direction, 10% chance to stay still, and with probability of 20% the agent moves in the opposite direction. We use the following policy in our experiments: (0, 1, 1, 0, 1, 1, 0, 0, 0, 0, 0, 0, 0, 0, 0, 0, 0, 0, 0, 0, 0, 0, 0, 0, 0, 1, 1, 1, 1, 1, 1, 1, 1, 1, 1, 1, 1, 1, 1, 1, 1, 1, 1, 1, 1, 0, 1, 0, 1, 1). The environment is shown in Figure 6.

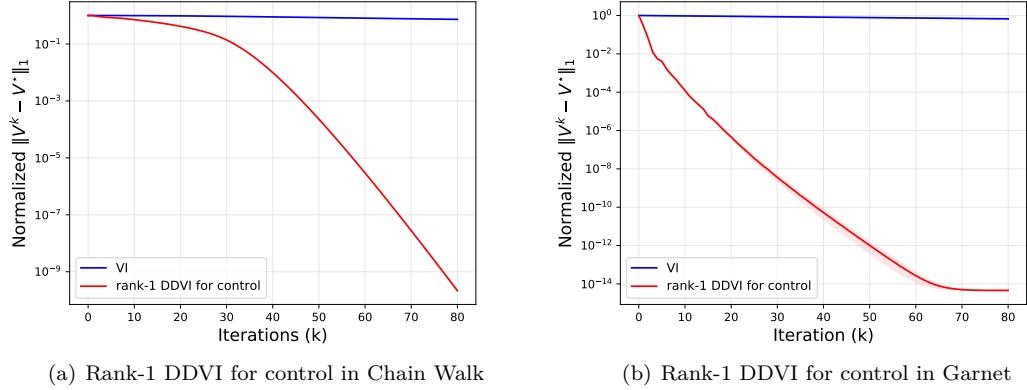

(a) Rank-1 DDVI for control in Chain Walk          (b) Rank-1 DDVI for control in Garnet

Figure 7: Comparison of rank-1 DDVI for control and VI in (left) Chain Walk and (right) Garnet.

**Garnet**  We use the Garnet environment as described by Farahmand & Ghavamzadeh (2021); Rakhsha et al. (2022), which is based on Bhatnagar et al. (2009). Garnet is parameterized by the tuple $(|\mathcal{X}|, |\mathcal{A}|, b_p, b_r)$. $|\mathcal{X}|$ and $|\mathcal{A}|$ are the number of states and actions, and $b_p$ is the branching factor of the environment, which is the number of possible next-states for each state-action pair. We randomly select $b_p$ states without replacement and then, transition distribution is generated randomly. We select $b_r$ states without replacement, and for each selected $x$, we assign state-dependent reward to be a uniformly sampled value in $(0, 1)$.

# E   Rank-$1$ DDVI for Control Experiments

In this experiment, we consider Chain Walk and Garnet. We use Garnet with 100 states, 8 actions, a branching factor of 6, and 10 non-zero rewards throughout the state space. We use normalized errors $\|V^k - V^\pi\|_1 / \|V^\star\|_1$, $\gamma = 0.995$, and $E_1 = \frac{1}{n}\mathbf{1}\mathbf{1}^\top$, where $n$ is the number of states. For Garnet, we plot average values on the 100 Garnet instances and denote one standard error with the shaded area. Figure 7(a) and 7(b) show that rank-1 DDVI for Control does indeed provide an acceleration.

# F Additional DDVI Experiments and Details

In all experiments, we set DDVI's $\alpha = 0.99$. We perform further experiments of DDVI with different ranks for discount factor $\gamma = 0.95, 0.995$ (Figure 8 and 9), and the QR iteration is run for 100 iterations in experiments.

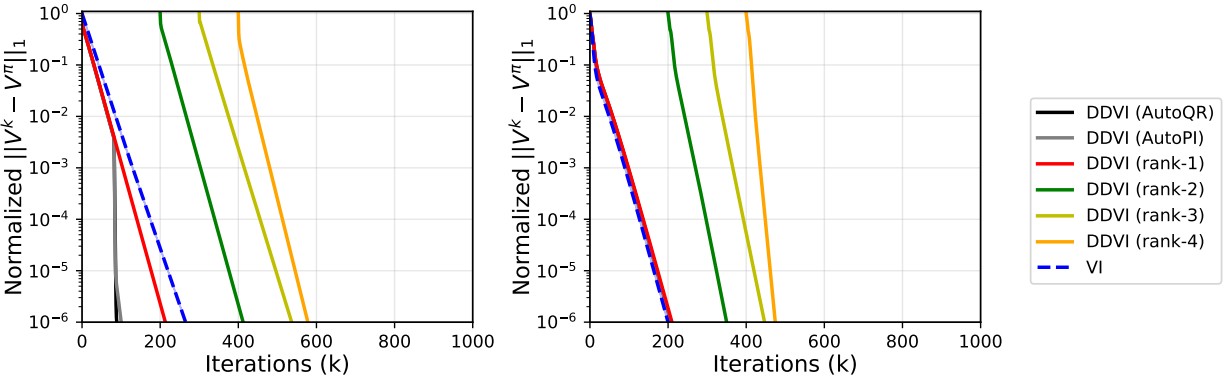

Figure 8: Comparison of DDVI with different ranks, AutoPI, and AutoQR against VI with discount factor $\gamma = 0.95$ in (left) Maze and (right) Chain Walk. The plots for DDVI do not start at iteration 0 because the costs of computing $E_s$ through QR iterations are incorporated as rightward shifts. Rate of DDVI with AutoPI and AutoQR changes when $E_s$ is updated.

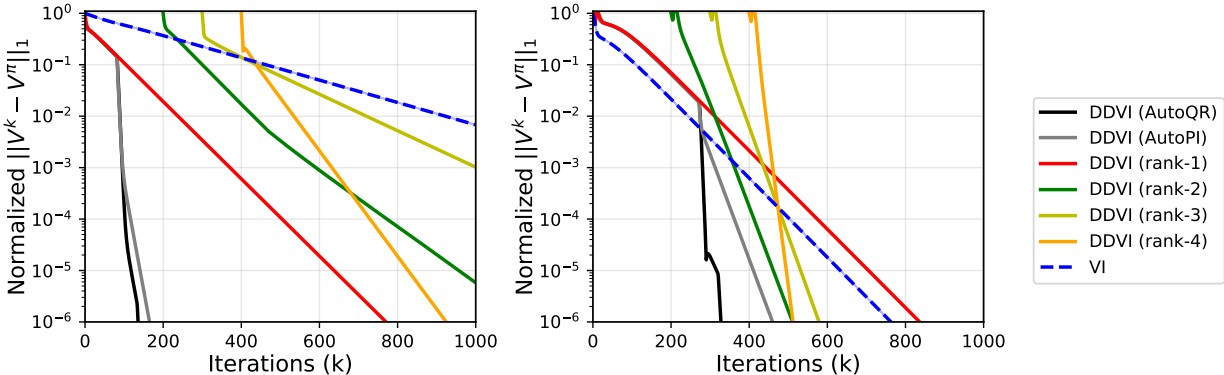

Figure 9: Comparison of DDVI with different ranks, AutoPI, and AutoQR against VI with discount factor $\gamma = 0.995$ in (left) Maze and (right) Chain Walk. The plots for DDVI do not start at iteration 0 because the costs of computing $E_s$ through QR iterations are incorporated as rightward shifts. Rate of DDVI with AutoPI and AutoQR changes when $E_s$ is updated.

We perform further comparison of convergence in Figures 10, 11, 12, 13. Figure 10 is run with Garnet environment with 50 states and 40 branching factor that matches the setting in Goyal & Grand-Clément (2022). In experiments, the QR iteration is run for 600 iterations. For PID VI, we set $\eta = 0.05$ and $\epsilon = 10^{-10}$. In Anderson VI, we have $m = 5$. In Figure 2, we use 20 Garnet MDPs with branching factor $b_p = 2$, and $b_r = 0.1|\mathcal{X}|$.

# G DDTD Experiments

A key part of our experimental setup is the model $\hat{\mathcal{P}}$. To show the robustness of DDTD to model error and also its advantage over Dyna, we consider the scenario that $\hat{\mathcal{P}}$ has some non-diminishing error. We achieve this with the same technique as Rakhsha et al. (2022). Assume that each iteration $t$, the empirical distribution of the next-state from $x, a$ is $\mathcal{P}_{\mathrm{MLE}}(\cdot|x, a)$, which is going to be updated with every sample. For

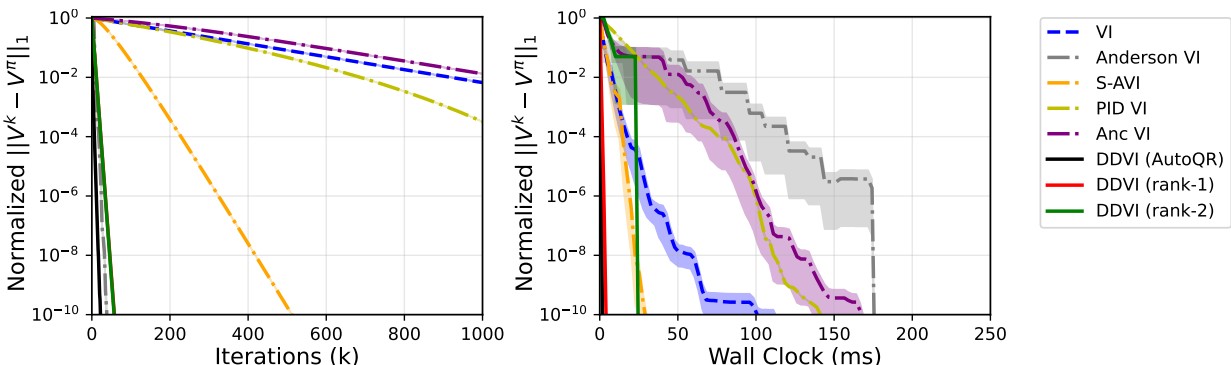

Figure 10: Comparison of DDVI with other accelerated VIs. Normalized errors are shown against the iteration number (left) and wall clock time (right). Plots are average of 20 randomly generated Garnet MDPs with 50 states and branching factor of 40 with shaded areas showing the standard error.

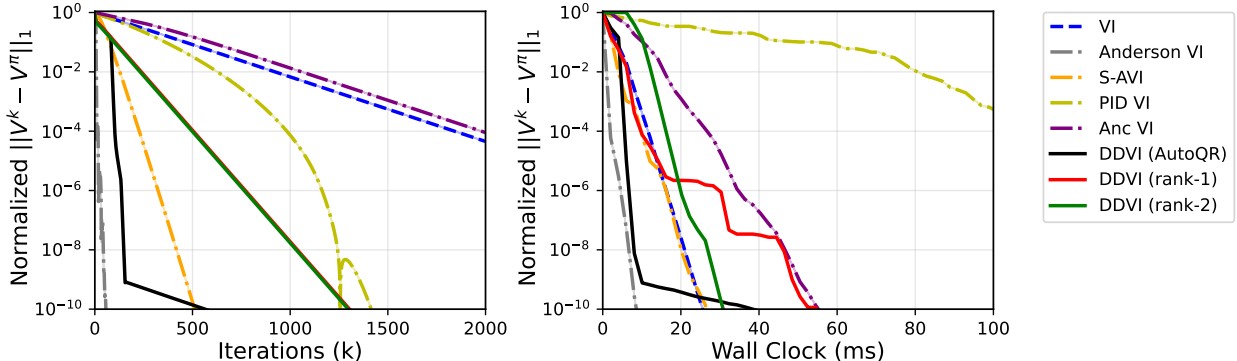

Figure 11: Comparison of DDVI with other accelerated VIs. Normalized errors are shown against the iteration number (left) and wall clock time (right) in Maze.

some hyperparameter $\theta \in [0, 1]$, we set the approximate dynamics $\hat{\mathcal{P}}(\cdot|x, a)$ as

$$\hat{\mathcal{P}}(\cdot|x, a) = (1 - \theta) \cdot \mathcal{P}_{\mathrm{MLE}}(\cdot|x, a) + \theta \cdot U(\{x'|\mathcal{P}_{\mathrm{MLE}}(x'|x, a) > 0\}) \tag{17}$$

where $U(S)$ for $S \subseteq \mathcal{X}$ is the uniform distribution over $S$. The hyperparameter $\theta$ controls the amount of error introduced in $\hat{\mathcal{P}}$. If $\theta = 0$, we have $\hat{\mathcal{P}} = \mathcal{P}_{\mathrm{MLE}}$ which becomes arbitrarily accurate with more samples. With larger $\theta$, $\hat{\mathcal{P}}(\cdot|x, a)$ will be smoothed towards the uniform distribution more, which leads to a larger model error. In Dyna, we keep the empirical average of past rewards for each $x, a$ in $\hat{r} : \mathcal{X} \times \mathcal{A} \to \mathbb{R}$ and perform planning with $\hat{\mathcal{P}}, \hat{r}$ at each iteration to calculate the value function, that is $V = (I - \gamma \hat{\mathcal{P}}^\pi)^{-1} \hat{r}^\pi$. In Figure 3, we have shown the result for $\theta = 0.3$. In Figure 14 and Figure 15 we show the results for $\theta = 0.1, 0.3, 0.5$ in both Maze and Chain Walk environments.

As we see in Figure 14 and Figure 15, DDTD shows a faster convergence than the conventional TD. In Maze, this is only achieved with higher rank versions of DDTD but in Chain Walk, even rank-1 DDTD is able to significantly accelerate the learning. Also note that unlike Dyna, DDTD is converging to the true value function despite the model error. We observe that the impact of model error on DDTD is very mild. In some cases such as rank-3 DDTD in Maze, between $\theta = 0.3$ and $\theta = 0.5$, we observe slightly slower convergence with higher model error. The hyperparamaters of TD Learning and DDTD are given in Tables 1, 2, and 3.

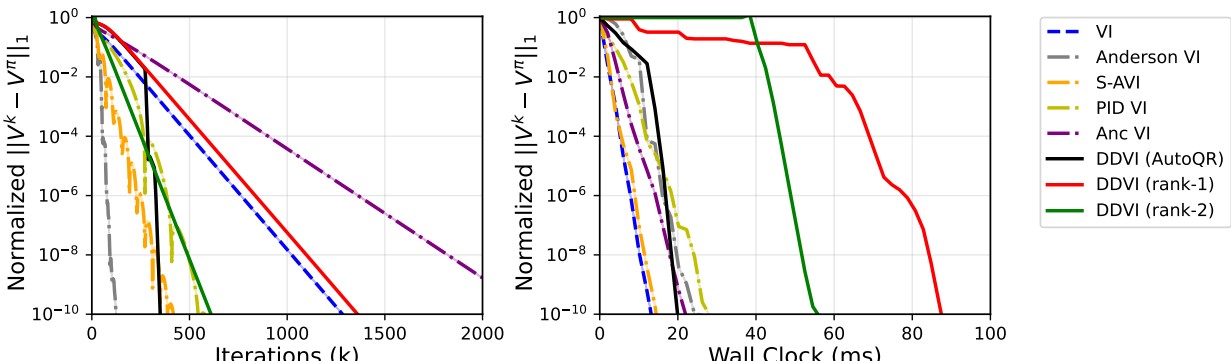

Figure 12: Comparison of DDVI with other accelerated VIs. Normalized errors are shown against the iteration number (left) and wall clock time (right) in Chain Walk.

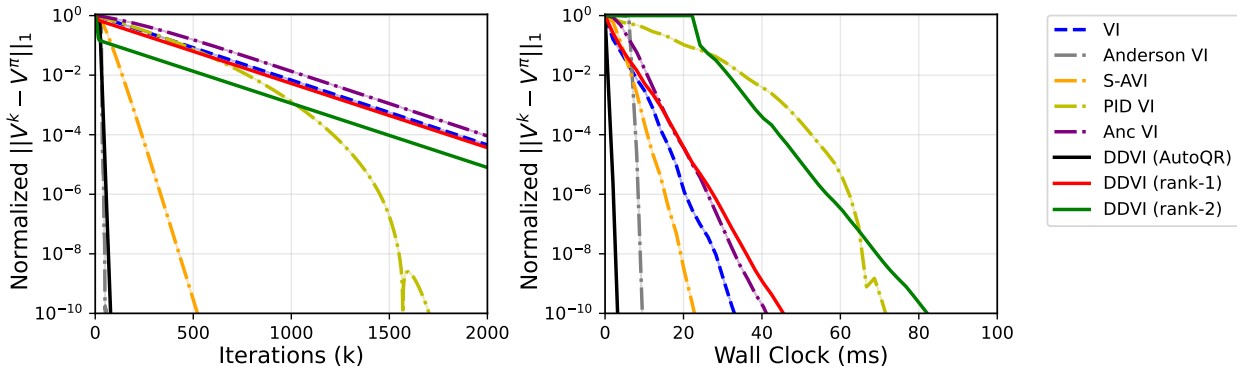

Figure 13: Comparison of DDVI with other accelerated VIs. Normalized errors are shown against the iteration number (left) and wall clock time (right) in Cliffwalk.

Table 1: Hyperparamters for the Maze environment for $\gamma = 0.99$. Cells with multiple values provide the value of the hyperparameter for $\theta = 0.1$, $\theta = 0.3$, and $\theta = 0.5$, respectively.

|  | rank-1 DDTD | rank-2 DDTD | rank-3 DDTD | rank-4 DDTD | TD |
|---|---|---|---|---|---|
| $\eta$ (learning rate) | $0.3, 0.3, 0.3$ | $0.3, 0.3, 0.3$ | $0.07, 0.07, 0.07$ | $0.07, 0.07, 0.07$ | $0.3$ |
| $\alpha$ | $0.8, 0.8, 0.8$ | $0.7, 0.8, 0.8$ | $0.9, 0.9, 0.9$ | $0.9, 0.9, 0.9$ | - |
| $K$ | 10 | 10 | 10 | 10 | - |

Table 2: Hyperparamters for the Maze environment for $\gamma = 0.95$. Cells with multiple values provide the value of the hyperparameter for $\theta = 0.1$, $\theta = 0.3$, and $\theta = 0.5$, respectively.

|  | rank-1 DDTD | rank-2 DDTD | rank-3 DDTD | rank-4 DDTD | TD |
|---|---|---|---|---|---|
| $\eta$ (learning rate) | $0.5, 0.5, 0.5$ | $0.5, 0.5, 0.5$ | $0.4, 0.4, 0.4$ | $0.4, 0.4, 0.4$ | $0.5$ |
| $\alpha$ | $0.8, 0.8, 0.8$ | $0.8, 0.8, 0.8$ | $0.8, 0.8, 0.8$ | $0.8, 0.8, 0.8$ | - |
| $K$ | 10 | 10 | 10 | 10 | - |

Table 3: Hyperparamters for the Chainwalk environment for both $\gamma = 0.99$ and $\gamma = 0.95$. Cells with multiple values provide the value of the hyperparameter for $\theta = 0.1$, $\theta = 0.3$, and $\theta = 0.5$, respectively.

|  | rank-1 DDTD | rank-2 DDTD | rank-3 DDTD | rank-4 DDTD | TD |
|---|---|---|---|---|---|
| learning rate ($\eta$) | 1 | 1 | 1 | 1 | 1 |
| $\alpha$ | $0.9, 0.9, 0.9$ | $0.9, 0.8, 0.8$ | $0.9, 0.8, 0.8$ | $0.9, 0.8, 0.8$ | - |
| $K$ | 10 | 10 | 10 | 10 | - |

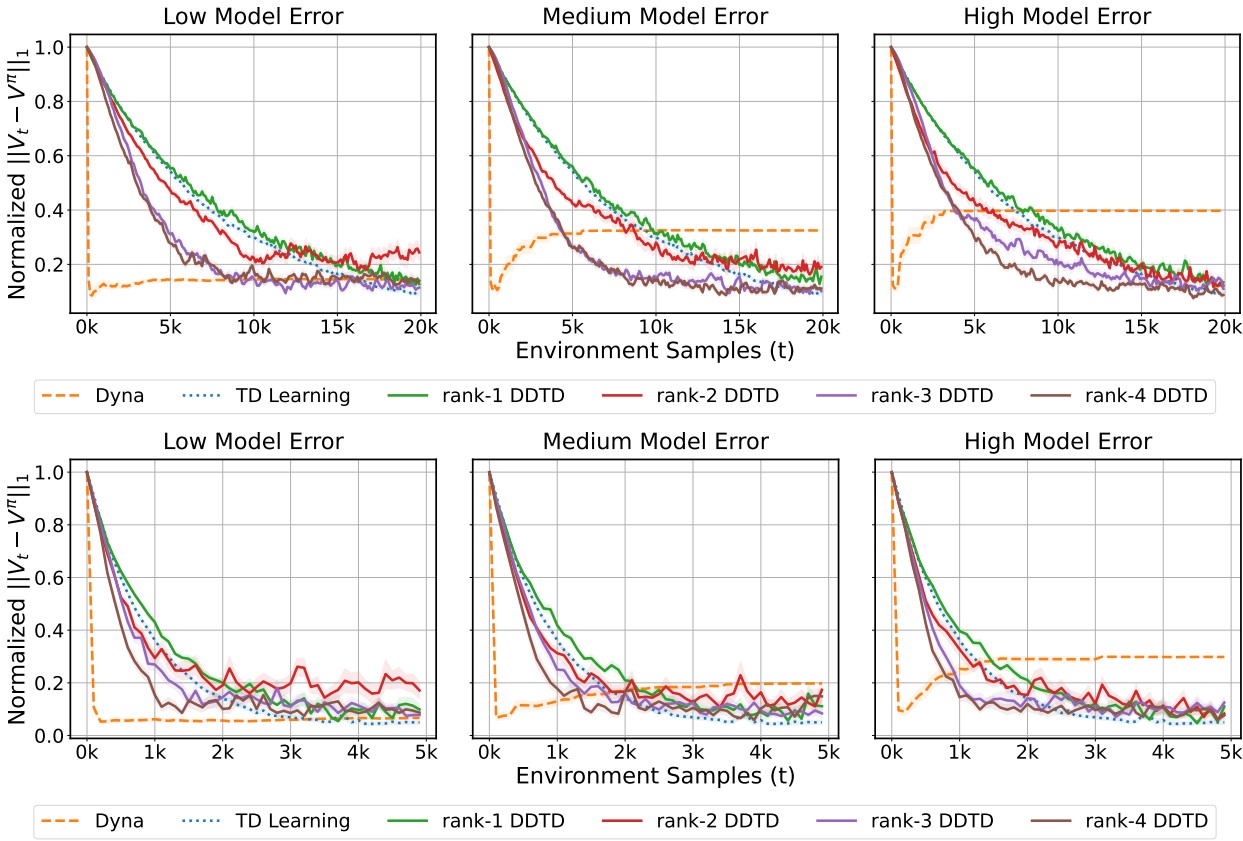

Figure 14: Comparison of DDTD with TD Learning and Dyna in Maze with $\gamma = 0.99$ *(Top)* and $\gamma = 0.95$ *(Bottom). Left:* low model error with $\theta = 0.1$. *Middle:* medium model error with $\theta = 0.3$. *Right:* high model error with $\theta = 0.5$. Each curve is average of 20 runs. Shaded area shows one standard error.

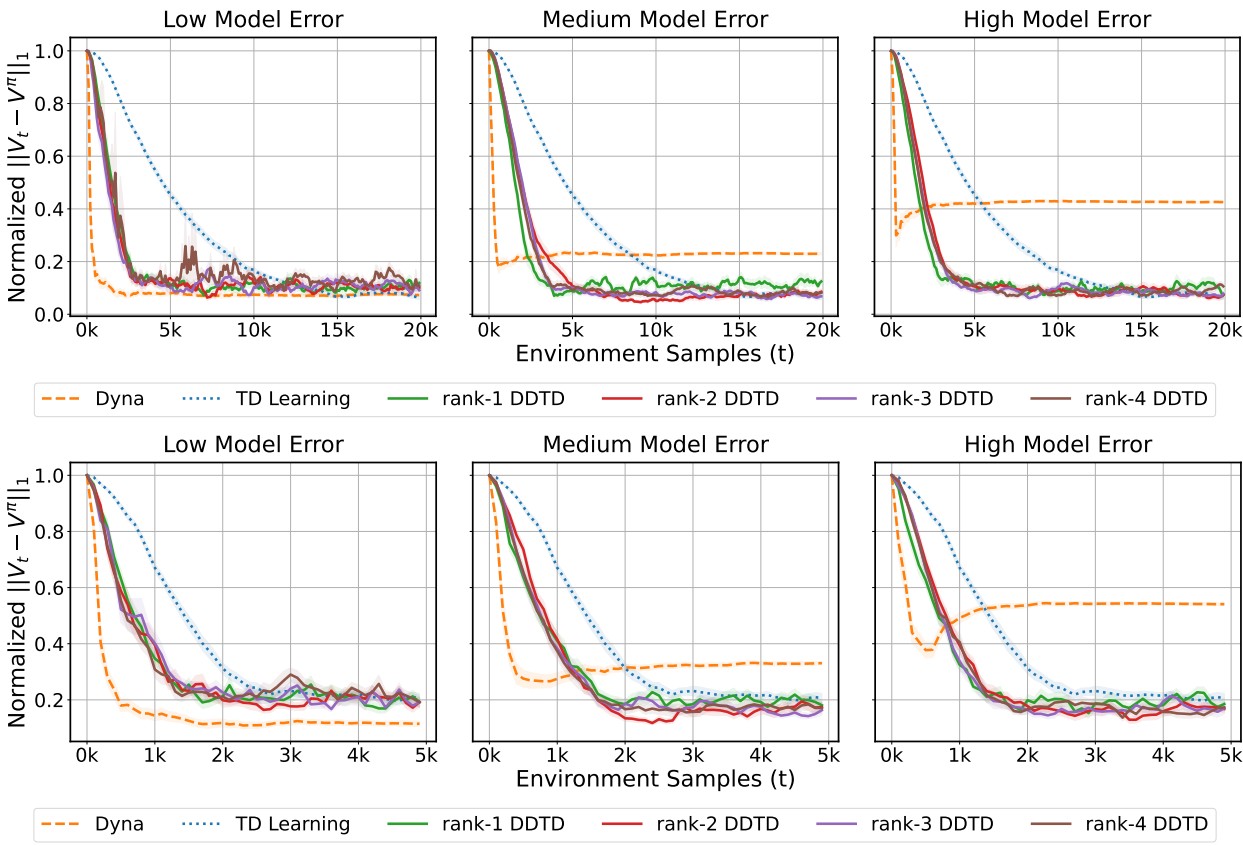

Figure 15: Comparison of DDTD with TD Learning and Dyna in Chainwalk with $\gamma = 0.99$ *(Top)* and $\gamma = 0.95$ *(Bottom)*. *Left:* low model error with $\theta = 0.1$. *Middle:* medium model error with $\theta = 0.3$. *Right:* high model error with $\theta = 0.5$. Each curve is average of 20 runs. Shaded area shows one standard error.

