# OpenReview forum: "Deflated Dynamics Value Iteration"
_TMLR — Accepted by TMLR_

### Review · Reviewer_xBYa · 2024-10-23

**Summary Of Contributions:**

This paper studies a fundamental problem of solving MDPs. It introduces the idea of matrix splitting and matrix deflation to effectively remove (deflate) the top s dominant eigen-structure of the transition matrix and proposes the Deflated Dynamics Value Iteration method. The paper shows that the convergence rate can be guranteed by $\gamma^k |\lambda_{s+1}|^k$, achieveing an acceleration of $|\lambda_{s+1}|^k$ compared with the traditional VI method. The paper also proposes a Deflated Dynamics Temporal Difference algorithm based on this offline value iteration method. Existensive experiments are conducted to verify the advantage of DDVI and DDTD compared with traditional VI and TD.

**Audience:**

Yes

**Claims And Evidence:**

Yes

**Requested Changes:**

Please see the last part.

**Strengths And Weaknesses:**

Strength:

The paper proposes a new value iteration methd based on matrix splitting and matrix deflation technique, and a new sample method based on this value iteration idea. The theoretical guarantees for this two methods are provided. For the former one, the authors show it achieves an $|\lambda_{s+1}|^k$ acceleration compared with the convergence rate of traditional VI.

The paper conduct rich experiments, comparing different variants of DDVI and other variants of VI for acceleration, in environments with different numbers of states and different discounting levels.

Weakness and questions:

In corollary 3.2, it shows that the convergence rate is $|\gamma \lambda_{s+1}|^k$ when $k$ tends to be $\infty$. However, for the classic VI method, the convergence rate does not require the iteration time to be $\infty$. Even though $\lambda$ may tend to be $1$, the infinite number of iterations of VI can also guarantee great convergence. Could you provide a discussion with respect to this point?

In Section 1, the authors mentioned some variants of VI to accelerate the convergence. More discussions and comparisons about their convergence rate should be provided.

Line 4 in page 3, a typo $X_k$.

In Section 5, it is unclear about the theoretical advantage of DDTD's convergence compared with that of TD.

In experiments, why selecting different discounting levels in experiments of DDTD and other experiments? The paper only compares the performances when $\lambda$ is very lage (larger than $0.99$). Other levels of $\lambda$ should also be tested and compared.

---

> ### Author Response · Authors · 2025-02-03
> **Response by authors**
>
> We thank the reviewer for detailed feedback.
>
> 1. As reviewer pointed out, the classic VI exhibits $O(\gamma^k)$-rate for any $k \ge 0$ and this rate can be derived using $l_\infty$-norm analysis, which is based on the the fact that $\||\gamma P^{\pi}V\|| \le \gamma \||V\|| $ for arbitrary policy $\pi$ and transition matrix $P$. On the other hand, the convergence rate of DDVI relies on eigenvalue analysis, which is based on fact that $\|A\|^k = \tilde{O}(\rho(A)^k)$ as $k \rightarrow \infty$, where $\rho(A)$ is spectral radius of $A$. Thus, even if $\rho(P^{\pi})=1$, classic VI achieves $O(\gamma^k)$-rate due to $l_\infty$-norm analysis, whereas spectral analysis yields $\tilde{O}(\gamma^k)$-rate.
>
> 2.  Due to space limitations, we defer the discussion of prior works to the Appendix. In the `Acceleration in RL' paragraph, we demonstrate variants of VI, including the methods used in our experiments. However, we acknowledge the reviewer's point that a more detailed discussion of their convergence rates should be provided. Regarding the convergence rate variants of VI used in our experiments, S-AVI [1] exhibits $O(\gamma_s^k)$-rate where $\gamma_s$ is a parameter satisfying $ \gamma \le \gamma_s \le 1$, PID VI [2] exhibits $O\left(\left(\frac{\sqrt{1+\gamma}-\sqrt{1-\gamma}}{\sqrt{1+\gamma}+\sqrt{1-\gamma}}\right)^k\right)$-rate under reversible MDP, with error dynamics analyzed for general MDPs, Anchored VI [3] exhibits $O\left(\frac{(\gamma^{-1}-\gamma)(1+2\gamma-\gamma^{k+1})}{(\gamma^{k+1})^{-1}-\gamma^{k+1}}\right)$-rate in terms of Bellman error, and Anderson VI framework [4] does not provide theoretical convergence rate. In our revised version, we updated the discussion of convergence rates of prior methods.
>
>
> 3. Thank you for pointing out typo. We updated typo and reflected in the revised version.
>
> 4.  In Section 5, Theorem 5.1 establishes the almost sure convergence of DDTD, and Theorem 5.2 demonstrates that higher rank DDTD allow for a larger range of convergent step sizes compared to the plain TD learning. However, we acknowledge that our theorems do not explicitly address the  acceleration of convergence for DDTD observed in the experiments, as Theroem 3.1 does in the DDVI setup. We plan to verify the theoretical accelerated convergence rate of DDTD in future work.
>
> 5. Reviewer raised valid point. Following reviewer's suggestion, we unified discount factor to $0.99$ for DDVI with different ranks and DDTD experiments in maintext. Additionally, we conducted further experiments with discount factors of $0.95, 0.99, 0.995$. We updated results in the Section F and G of Appendix of the revised version and observed that DDVI and DDTD consistently showed valid acceleration.
>
>
>
> [1] V. Goyal and J. Grand-Clément. A first-order approach to accelerated value iteration. Operations Research,
> 71(2):517–535, 2022.
>
> [2] A. Farahmand and M. Ghavamzadeh. PID accelerated value iteration algorithm. International Conferenceon Machine Learning, 2021.
>
> [3] J.Lee and E. K. Ryu. Accelerating value iteration with anchoring. Neural Information Processing Systems, 2023.
>
> [4] M. Geist and B. Scherrer. Anderson acceleration for reinforcement learning. European Workshop on Reinforcement Learning, 2018.

---

### Review · Reviewer_rh77 · 2024-10-26

**Summary Of Contributions:**

This paper proposed to improve the standard value iteration which has convergence rate $O(\gamma^k)$, when $\gamma \approx 1$, the convergence rate can be slow as have been observed by existing work. This paper built on the deflation matrix technique and proposed numerical stable solutions. By removing a matrix from the transition dynamics matrix, it is possible to accelerate the convergence rate up to $\tilde{O}(\gamma^k |\lambda_{s+1}^k|)$ which can be significantly faster.

**Audience:**

Yes

**Broader Impact Concerns:**

ethical concerns not applicable

**Claims And Evidence:**

Yes

**Requested Changes:**

I don't see major issues of the paper. However, I would like to ask the authors to provide additional discussion on how the proposed method can be used to accelerate popular VI methods like DQN, which was briefly discussed in the beginning of section 5. In my understanding, DDTD is an on-policy algorithm, which makes it hard to compete against Q-learning/DQN that can reuse samples. However, as the authors pointed out, slow convergence has been observed for these algorithms. It would be very beneficial to have a paragraph on this point on how DDTD could extend to these cases.

**Strengths And Weaknesses:**

The paper is very well-written. The authors did an excellent job in positioning the paper, explaining the issue with value iteration, presenting technical details in the preliminary and the contribution sections. The theoretical results seem convincing and to the point of the paper.

---

> ### Author Response · Authors · 2025-02-03
> **Response by authors**
>
> We are happy to hear that the reviewer found no major issues in our paper.
>
> Reviewer raised good point. We believe that our DDTD framework could be potentially extended to Deep Q-Network (DQN) setting. Here, we outline a rough sketch of a DQN-based version of DDTD as follows:
>
>
> 1. Sample $\\{(X_i, A_i, R_i, X_i')\\}^N_{i=1}$ by following policy $\pi$ and add the samples to replay buffer.
>
> 2.  Approximate deflation matrix $E_s$ of $P^{\pi}$ using samples in replay buffer.
>
> 3. For $k=1,2, \dots$
>
> (1) Update parameters:
>
> $\theta_k=argmin_{\theta}\frac{1}{N}\sum^N_{i=1}\||R_i+\gamma V^k(X'_i)-\gamma E_s V^k (X_i)-W(\theta) (X_i, A_i)\||^2$
>
>
> (2) Update the value function:
>
> $V^{k+1} = (I-\gamma E_s)^{-1}W(\theta_k).$
>
> Note that $W(\theta_k)$ is represented by a function approximator, such as a deep neural network (DNN) and $V^k$ is estimation of value function. We expect that Step 3-(1) could be handled similarly to the DQN framework. In our view, the most challenging part of this framework lies in Step 2, where deflation matrix $E_s$ is approximated. In the finite state case, as in our DDTD framework, we can learn the model $\hat{P}^{\pi}$ from samples in replay buffer and run QR-iteration to derive $E_s$. For continuous state spaces, however, it remains an open question how to approximate $E_s$, as it is an operator defined based on the eigen-functions. We consider this an interesting direction for future research and included this discussion in conclusion Section of updated version.

---

### Review · Reviewer_avGa · 2025-01-20

**Summary Of Contributions:**

The authors propose a matrix splitting technique in the context of value iteration and temporal difference learning ,and establish that it rigorously improves the convergence rate relative to prior art. Experiments demonstrate the merits of the proposed approach.

**Audience:**

Yes

**Claims And Evidence:**

Yes

**Requested Changes:**

Minor comment: SOR should be given as non-acronym (Successive over-relaxation) for subsection title of 3.2, since it is first introduced inside the subsection. Is the specific form of SOR presented in section 3.2 with A=B+C+D unique to this work, or appears in prior art? If it is the former, could the authors clarify why this generalized decomposition is important or required for the policy evaluation setting?

Also, several of the quantities specified here are for the space of complex numbers/matrices, but I do not believe any of these probability matrices are actually complex-valued but instead real-valued.

For the main presentation of equation (2), the authors should contrast the fixed point iteration with the fixed point operator that is usually applied for value iteration. How is the proposal a special variant of a Bellman operator that is restricting focus t the eigenspace of the transition matrix? Not enough time is spent interpreting the meaning of the core algorithm derivation in Section 3.3.

For the convergence rate comparison with vanilla value iteration, the authors should include dependence on the parameter dimension that also clarifies the computational cost required to evaluate the eigenvalue decompositions proposed. Without this, reporting the pure acceleration of the rate is not valid/fair.

A similar comment holds for the QR decomposition, as well as comparison with vanilla temporal difference iteration.

Experimentally, the authors should compare with alternative techniques for variance reduction, such as SAG, SVRG, zap-Q learning (stochastic Newton Raphson, etc.) to make the case that the specific class of power iteration methods in eigenspace have merits relative to alternative notions of acceleration/variance reduction. Are there specific classes of transition matrices that are particularly advantageous for the proposed algorithm?

**Strengths And Weaknesses:**

See below.

---

> ### Author Response · Authors · 2025-02-03
> **Response by Authors**
>
> We thank the reviewer for constructive feedback.
>
> 1. Thank you for pointing our mistake. As reviewer suggested, we changed the subsection title of 3.2 to `Successive Over-Relaxation' in revised version.
>
>
> 2. Classical SOR has the form $A=B+C+D$, which assumes that $B,C,D$ are a lower triangular, upper triangular, and diagonal matrix, respectively. In our DDVI framework, we generalizes this framework and use $b=r^{\pi}, B=-\gamma E, C=\gamma (P^{\pi}-E), D=I$ (Note that there is no guarantee that our choices of $B,C,D$ satisfy assumptions of classical SOR). This generalized decomposition was necessary and important for combining the operator splitting framework with the deflation matrix technique as we demonstrated in Section 3 .
>
>
> 3. As the reviewer pointed out, probability matrices are real-valued. However, our discussion in Section $3.1$ focuses on eigenvalues, which can be complex. Even in such cases, prior deflation techniques have successfully deflated dominant eigenvalues.
>
>
> 4. Sorry for the confusion. Before presenting equation (2), we have added the fixed-point iteration form of DDVI to improve understanding in the revised version. In the previous subsection, we clarified that value iteration can be derived from our SOR iteration by setting $\alpha = 1$ and $E = 0$, demonstrating that value iteration is a specific instance of DDVI.
>
> 5. The reviewer raised a valid issue. In the revised version, we include a plot (Figure 1  in Section $6$) reflecting the computational cost of QR iteration. Specifically, we consider the cost of a QR iteration per step to be equivalent to that of value iteration, as in both cases, the dominant computational cost arises from product between transition matrix and iterate. So the cost of a QR iteration is $msC$ where $m$ is the number of QR iterations we applied, $s$ is the rank of DDVI, and $C$ is the cost of VI per iteration, and we plot rank-$s$ DDVI starting from $ms$ iterations. In contrast, in the sampling-based setup, we assume that the dominant cost arises from sampling, and in the DDTD framework, QR iteration is applied to the learned model, which does not involve sampling process. For this reason, we chose to reflect solely the sampling cost in our plot.
>
> 6. Thank you for your suggestion. Comparing existing variance reduction methods with our DDTD framework in a sampling-based setup would be an interesting work. However, we believe it is appropriate to leave this work as future work, as it would take time to determine how these algorithms can be applied to the policy evaluation problem and to establish proper criteria for comparison with our DDTD method. For instance, as far as we know, SAG and SVRG are well-known variance reduction methods in optimization literature. Specifically, they are stochastic gradient descent methods with variance reduction sampling designed for minimizing a finite sum of functions, while DDTD solves the Bellman equation, fixed-point problem. The reviewer may consider reformulating the policy evaluation problem from the Bellman equation into a bilinear saddle-point formulation and then applying SVRG or SAG to the corresponding primal or dual problem. However, this would require some knowledge from another field, extend beyond the scope of our framework, and necessitates clear criteria for comparing methods from different domains. Thus, we leave this comparison for future research.
>
> 7. The reviewer raised a interesting point. Yes, DDVI would perform efficiently for MDPs with sparse transition matrices. As our our main presentation equation $2$ shows, the dominant cost of DDVI, $O(sn^2)$, comes from matrix-vector multiplication $P^{\pi}V$. Since sparse matrices have a significant number of zero elements, the cost of matrix multiplication and addition are reduced efficiently [2, Section 2] and this lead to efficient cost of DDVI. Also, there is extensive research on finding eigenvalues of large sparse matrices [1, Section 10; 2, Section 8]. Among these, "Sparse QR factorization" [2, Section 5.2] adapts QR factorization to large sparse matrices, with complexity depending on the sparsity pattern. Therefore, with sparse QR factorization, we expect that DDVI perform more computationally efficient.
>
> [1] G. H. Golub and C. F. Van Loan. Matrix Computations. The John Hopkins University Press, 4th edition, 2013
>
> [2] Davis, T. A. (2006). Direct methods for sparse linear systems. Society for Industrial and Applied Mathematics.

---

### Decision · Action_Editor_iAw2 · 2025-03-05

**Recommendation:** Accept as is

**Comment:**

The reviewers agree on the quality of the submission. In particular they highlight that the matrix splitting technique is a particularly interesting contribution, that the paper is well written, that the paper is well positioned in the related literature, and that the experiments are complete.
Only minor issues were raised, which have already been addressed by the authors in the revised manuscript.

**Audience:**

The findings of this paper are of interest to the planning and reinforcement learning communities.

**Claims And Evidence:**

The claims are supported by a complete theoretical analysis and corroborated by convincing experiments.